# MEAN FIELD LANGEVIN ACTOR-CRITIC: FASTER CONVERGENCE AND GLOBAL OPTIMALITY BEYOND LAZY LEARNING

## ABSTRACT

We study how deep reinforcement learning algorithms learn meaningful features when optimized for finding the optimal policy. In particular, we focus on a version of the neural actor-critic algorithm where both the actor and critic are represented by over-parameterized neural networks in the mean-field regime, and are updated via temporal-differece (TD) and policy gradient respectively. Specifically, for the critic neural network to perform policy evaluation, we propose *mean-field Langevin TD learning* method (MFLTD), an extension of the mean-field Langevin dynamics with proximal TD updates, and compare its effectiveness against existing methods through numerical experiments. In addition, for the actor neural network to perform policy updates, we propose *mean-field Langevin policy gradient* (MFLPG), which implements policy gradient in the policy space through a version of Wasserstein gradient flow in the space of network parameters. We prove that MFLTD finds the correct value function, and the sequence of actors created by MFLPG created by the algorithm converges linearly to the globally optimal policy of the Kullback Leibler divergence regularized objective. To our best knowledge, we provide the first linear convergence guarantee for neural actor-critic algorithms with *global optimality* and *feature learning*.

## 1 INTRODUCTION

In recent years, the field of reinforcement learning (RL) (Sutton & Barto, 2018) including the policy gradient method (Williams, 1992; Baxter et al., 1999; Sutton et al., 1999) and the temporal-difference (TD) learning (Sutton, 1988) has made tremendous progress, with deep reinforcement learning methods. The combination of the actor-critic method (Konda & Tsitsiklis, 1999) and neural networks has demonstrated significant empirical success in challenging applications, such as the game of Go (Silver et al., 2016; 2017) or the human-like feedback alignment (Ouyang et al., 2022). In these empirical successes, the employment of deep neural networks plays an indispensable role — their expressivity enable learning meaningful features that benefit decision-making. However, despite the impressive empirical results, there remain many open questions about the theoretical foundations of these methods. In particular, when viewing deep RL methods as optimization algorithms in the space of neural network policies, it remains elusive how deep RL algorithms learn features during the course of finding the optimal policy.

One source of difficulty in the analysis of neural policy optimization comes from the nonconvexity of the expected total reward over the policy space. Also, TD learning used in the policy evaluation subproblem faces classic challenges (Baird, 1995; Tsitsiklis & Van Roy, 1996) stemming from the bias of semi-gradient optimization (Sutton, 1988). Another source of difficulty is the nonlinearity associated with the neural networks parameterizing both the policy and state-action value functions. The tremendous success of deep RL is attributed to its rich expressive power, which is backed by the nonlinearity of neural networks, which at the same time brings a considerable challenge to the optimization aspect. Unfortunately, the advantages of data-dependent learning of neural networks in the context of RL have only a limited theoretical understanding. Classical theoretical studies of policy optimization and policy evaluation problems, including the actor-critic method, limit their analysis to the case of linear function approximation in both the actor and the critic, where the feature mapping is fixed during learning (Sutton et al., 1999; Kakade, 2001; Bhatnagar et al., 2007; 2009). Recently, some analyses based on the theory of Neural Tangent Kernel (NTK) (Jacot et al., 2018) are established, which state that an infinite-width neural network is well approximated by a linear

function of random features determined by initial parameters under certain conditions (Cai et al., 2019; Wang et al., 2020; Liu et al., 2019). More recent works (Zhang et al., 2020; 2021) establish the study of convergence and optimality of over-parameterized neural networks over lazy training (Chizat et al., 2019), incorporating a mean-field perspective corresponding to NTK. Specifically, by letting the network width be sufficiently large under appropriate conditions in NTK or lazy training regimes, optimality is guaranteed based on the fact that the neural network features are as close as possible to the data-independent initial feature representation. In other words, these existing analyses do not fully capture the representation learning aspect of neural RL empowered by the expressivity of neural networks. Thus, in this paper, we aim to address the following question:

*Does neural actor-critic provably learn features on the way to the global optima?*

We provide an affirmative answer to this question by focusing on the case where both the actor and the critic are represented by an over-parameterized two-layer neural network in the mean-field regime. Under this setting, we propose to update the actor and critic by a variant of policy gradient and TD learning tailored to mean-field neural networks, based on Langevin dynamics. We prove that the critic converges to the correct value function sublinearly and the sequence of actors converges to the globally optimal policy of a Kullback Leibler (KL) divergence regularized objective. More importantly, our theory is beyond the lazy training regime and provably shows that the actor and critic networks performs feature learning in the algorithm.

**Our Contributions** The main contribution of this paper is to propose the Mean-field Langevin actor-critic algorithm and prove linear convergence and global optimality with *feature learning* (Suzuki, 2019; Ghorbani et al., 2019). We treat the problem of policy improvement and policy evaluation as an optimization over a probability distribution of network parameters with KL-divergence regularization and build convergence analysis based on *mean field Langevin dynamics* (MFLD). Specifically,

1. We introduce the *mean-field Langevin TD learning* (MFLTD) as the policy evaluation component (critic) and show that it converges to the true value function at a sublinear rate. In this algorithm, we employ a double-loop proximity gradient algorithm to resolve the difficulties posed by having semi-gradients instead of gradients of the mean-square Bellman error in TD-learning.

2. We introduce the *mean-field Langevin policy gradient* (MFLPG) as the policy improvement component (actor) and prove that it converges to the globally optimal policy of expected total reward at a linear convergence rate under KL-divergence regularization. This algorithm is equivalent to the standard policy gradient in the parameter space with additional injected noises.

Our analysis extends the convergence analysis of MFLD with general over-parameterized neural networks (Nitanda et al., 2022; Chizat, 2022) to both TD learning and the policy gradient methods. At the core of our analysis are (1) the over-parameterization of two-layer neural networks to represent policies and approximate state-action value functions, (2) the strong convexity-like properties acquired by the objective function through KL-divergence regularization, (3) the proximal gradient algorithm for TD learning to prevent convergence breakdown by using the semi-gradient of the mean squared Bellman error, and (4) the use of geometric property taking advantage of the universal approximation of the Barron-like class to connect the convergence of the policy gradient method with the one-point convexity from Kakade & Langford (2002). In particular, (1) attributes the problem to the Wasserstein gradient flow and enables to utilize the convexity of the loss function in the measure space. Furthermore, together with (2), it induces the log-Sobolev inequality, which guarantees linear convergence speed in the presence of globally convergent solutions. Note here that, our whole results are valid with arbitrary regularization parameters. To the best of our knowledge, our analysis gives the first global optimality and linear convergence guarantees for the neural policy gradient methods with feature learning, confirming their considerable empirical success.

**Related Works** Regarding the convergence and optimality of the actor-critic, there is a need to encompass the two optimization problems of the actor component and the critic component, and in terms of the complexity of each problem, the theoretical research is limited. Regarding TD learning, various approaches mainly utilizing linear function approximation have been made to address the divergence and non-convergence issues arising from semi-gradient (Baird, 1995; Tsitsiklis & Van Roy, 1996). In particular, Capturing neural networks in the NTK regime, Cai et al. (2019) demonstrated sublinear convergence to the true value function, and Zhang et al. (2020) showed such sublinear convergence by attributing this optimization to lazy training. On the other hand, the global convergence of policy gradient methods is limited due to the non-convexity of the objective function, but Fazel et al. (2018); Yang & Wang (2019) proved the convergence of policy gradient methods to the globally optimal policy in the LQR setting (Fazel et al., 2018), and Bhandari & Russo (2019);

Agarwal et al. (2020) proved convergence to the globally optimal policy in tabular and their own linear settings. Along the line of research, Wang et al. (2020) incorporated Cai et al. (2019) as the critic component, assuming that both the actor and critic, over-parameterized neural networks, are well approximated by linear functions of random features determined by initial parameters. They provided convergence to the globally optimal policy at a sublinear rate. However, these analyses over NTK or lazy training regimes assume that the neural network does not learn features from the input data.

As opposed to the linearization analysis above, we use the following tools of mean-field Langevin theory. In general, gradient method analysis of mean-field neural networks uses the convexity of the objective in the space of probability measures to show its global optimality (Nitanda & Suzuki, 2017; Chizat & Bach, 2018; Mei et al., 2018), MFLD yields to an entropy regularization term in the objective by adding Gaussian noises to the gradient. Within this research stream, our work is closely related to Nitanda et al. (2022); Chizat (2022) using convex analysis focusing on the log-Sobolev inequality starting from the Nitanda et al. (2021). There is also a large body of literature analyzing the optimization analysis of supervised learning with over-parameterized neural networks in the mean-field regime (Hu et al., 2021; Chen et al., 2020; Nitanda et al., 2022; Chizat, 2022).

## 2 BACKGROUND

The agent interacts with the environment in a discounted Markov decision process (MDP) (Puterman, 2014) given by a tuple $(\mathcal{S}, \mathcal{A}, \gamma, P, r)$. The policy $\pi : \mathcal{S} \times \mathcal{A} \to \mathscr{P}(\mathcal{S})$ represents the probability at which the agent takes a specific action $a \in \mathcal{A}$ at a given state $s \in \mathcal{S}$, with the agent receiving a reward $r(s, a)$ when taking an action $a$ at state $s$, and transitioning to a new state $s' \in \mathcal{S}$ according to the transition probability $P(\cdot|s, a) \in \mathscr{P}(\mathcal{S})$. Rewards are received as an expected total reward $J[\pi] = \mathbb{E}[\sum_{\tau=0}^{\infty} \gamma^\tau r_\tau | a_\tau \sim \pi(s_\tau)]$, with $\gamma \in (0, 1)$ being the discount factor.

Here, we denote the state-value function and the state-action value function (Q-function) associated with $\pi$ by $V_\pi(s) = (1 - \gamma) \cdot \mathbb{E}\left[\sum_{\tau=0}^{\infty} \gamma^\tau \cdot r(s_\tau, a_\tau) \mid s_0 = s, a_\tau \sim \pi(s_\tau), s_{\tau+1} \sim P(s_\tau, a_\tau)\right]$ and $Q_\pi(s, a) = (1 - \gamma)\mathbb{E}\left[\sum_{\tau=0}^{\infty} \gamma^\tau \cdot r(s_\tau, a_\tau) \mid s_0 = s, a_0 = a, a_\tau \sim \pi(s_\tau), s_{\tau+1} \sim P(s_\tau, a_\tau)\right]$.

Note that policy $\pi$ with the transition kernel $P$ induces a Markov chain over state space $\mathcal{S}$, and we make the assumption that every policy $\pi$ is ergodic, i.e. has a well-defined stationary state distribution $\varrho_\pi$ and the stationary state-action distribution $\varsigma_\pi = \pi(a|s) \cdot \varrho_\pi(s)$. Moreover, we define the state visitation measure and the state-action visitation measure induced by policy $\pi$, respectively, as

$$\nu_\pi(s) = (1 - \gamma) \cdot \sum_{\tau=0}^{\infty} \gamma^\tau \cdot \mathbb{P}\left(s_\tau = s \mid a_\tau \sim \pi(s_\tau), s_{\tau+1} \sim P(s_t \tau, a_\tau)\right), \quad \sigma_\pi(s, a) = \pi(a|s) \cdot \nu_\pi(s),$$

which counts the discounted number of steps that the agent visits each $s$ or $(s, a)$ in expectation.

**Policy Gradient** Here, we define the expected total reward function $J[\pi]$ for all $\pi$ as

$$J[\pi] = (1 - \gamma) \cdot \mathbb{E}\left[\sum_{\tau=0}^{\infty} \gamma^\tau \cdot r(s_\tau, a_\tau) \,\middle|\, a_t \sim \pi(s_\tau), s_{\tau+1} \sim P(s_\tau, a_\tau)\right].$$

The goal of the policy gradient ascent is to maximize $J[\pi]$ by controlling policy $\pi$ under the reinforcement learning setting defined above, where the optimal policy is denoted by $\pi^*$. We parameterize the policy as $\pi_\theta$, where $\theta \in \Theta$ is the parameter. We aim to adjust the parameters of the policy in the direction of the gradient of the expected cumulative reward with respect to the parameters with some approximations. The gradient of $J[\pi_\Theta]$ over $\Theta$ is introduced by the policy gradient theorem (Sutton et al., 1999) as $\nabla_\Theta J[\pi_\Theta] = \mathbb{E}_{\nu_{\pi_\Theta}}\left[\int \nabla_\Theta \pi_\Theta(da|s) \cdot Q_{\pi_\Theta}(s, a)\right]$. The value function in the above gradient is estimated by the policy evaluation problem.

**Temporal-Difference Learning** In temporal-difference (TD) learning, we parameterize a Q-function as $Q_\Omega$ and aim to estimate $Q_\pi$ by minimizing the mean-squared Bellman error (MSBE):

$$\min_\Omega \text{MSBE}(\Omega) = \mathbb{E}_{\varsigma_\pi}\left[(Q_\Omega(s, a) - \mathcal{T}^\pi Q_\Omega(s, a))^2\right], \tag{1}$$

where $\mathcal{T}^\pi$ is the Bellman evaluation operator associated with policy $\pi$, which is defined by $\mathcal{T}^\pi Q(s, a) = \mathbb{E}\left[r(s, a) + \gamma Q(s', a') \mid s' \sim P(s, a), a' \sim \pi(s')\right]$, and $Q_\Omega$ is a Q-function parameterized with parameter $\Omega$. The most common example of TD-learning is TD(0) algorithm, which, in the population version, updates $\Omega$ via the semi-gradient $\mathbb{E}_{\varsigma_\pi}\left[(Q_\Omega(s, a) - \mathcal{T}^\pi Q_\Omega(s, a)) \cdot \nabla_\Omega Q_\Omega(s, a)\right]$.

## 3 MEAN-FIELD LANGEVIN POLICY GRADIENT

In this section, we introduce a particle-based double-loop neural actor-critic method with the policy and Q-function parameterized by neural networks in discrete time and the convergence analysis in the mean-field limit. We first introduce the parameterization of actor and critic below.

**Parameterization of Policy and Q-Function**   For notational simplicity, we assume that $\mathcal{S} \times \mathcal{A} \subset \mathbb{R}^D$ with $D \geq 2$ and that $\|(s,a)\| \leq 1$ for all $(s,a) \in \mathcal{S} \times \mathcal{A}$ without loss of generality. We parameterize a function $h : \mathcal{S} \times \mathcal{A} \to \mathbb{R}$ using a two-layer neural network with width $m$ and $d$-dimensional parameters $\Theta = (\theta^{(1)}, \ldots, \theta^{(m)}) \in \mathbb{R}^{d \times m}$ where it holds that $d = D + 2$, which is denoted by $\mathrm{NN}(\Theta; m)$,

$$f_\Theta(s,a) = \frac{1}{m} \sum_{i=1}^m h_{\theta^{(i)}}(s,a), \quad h_\theta(s,a) = R \cdot \beta(b) \cdot \sigma(w^\top(s,a,1)), \quad \theta = (w,b), \quad (2)$$

where $h_\theta(s,a) : \mathcal{S} \times \mathcal{A} \to \mathbb{R}$ is the nonlinear transformation function, $\sigma : \mathbb{R} \to \mathbb{R}$ is the activation function, $\beta : \mathbb{R} \to (-1,1)$ is a bounded function that represents the second layer weights with the bound $R > 0$. We now introduce the parameterization of the policy $\pi$ and the Q-function $Q$ with neural networks in the mean-field regimes respectively. Let $f_\Theta = \mathrm{NN}(\Theta; m), f_\Omega = \mathrm{NN}(\Omega; m)$. Then we denote the policy and Q-function by $\pi_\Theta$ and $Q_\Omega$, which are given by

$$\pi_\Theta(a|s) = \exp\left(-f_\Theta(s,a) - \ln Z_\Theta(s)\right), \qquad Q_\Omega(s,a) = f_\Omega(s,a),$$

where $Z_\Theta$ is a normalization term and, by the definition, we have $\int \pi_\Theta(a|s)\mathrm{d}a = 1$ for all $s \in \mathcal{S}$.

**Mean-field Limit**   By taking mean-field limit $m \to \infty$, we obtain the policy $\pi_\rho$ and the Q-function $Q_q$ induced by the weight distributions $\rho, q \in \mathcal{P}_2$, respectively,

$$\pi_\rho(a|s) = \exp\left(-\mathbb{E}_{\theta \sim \rho}[h_\theta(s,a)] - \ln Z_\rho(s)\right), \qquad Q_q(s,a) = \mathbb{E}_{\omega \sim q}[h_\omega(s,a)], \quad (3)$$

where $Z_\rho(s)$ is a normalization term making $\pi_\rho(\cdot|s)$ a probability distribution on $\mathcal{A}$. We now impose the following assumption on the two-layer neural network $h_\theta$.

**Assumption 1** (Regularity of the neural network.). *For the neural network $h_\theta$ defined in Eq. (2), we assume the activation function $\sigma : \mathbb{R} \to \mathbb{R}$ is uniformly bounded, $L_1$-Lipschitz continuous, and $L_2$-smooth. Besides, we assume the second weight function $\beta : \mathbb{R} \to (-1,1)$ is an odd function which is $L_3$-Lipschitz continuous and $L_4$-smooth.*

Without loss of generality, we can assume $\sigma \in (-1,1)$, which implies that the neural network $h_\theta$ is bounded by $R > 0$. Assumption 1 is a mild regularity condition except for the boundary of the neural network. Assumption 1 can be satisfied by a wide range of neural networks, e.g., $\beta(\cdot) = \tanh(\cdot/R)$ and $\sigma(\cdot) = \tanh(\cdot)$. We further redefine $J : \rho \mapsto J[\rho] := J[\pi_\rho]$ as a functional over $\rho$.

### 3.1 ACTOR UPDATE: MEAN-FIELD LANGEVIN POLICY GRADIENT

We aim to minimize the regularized negative expected total rewards $J[\rho]$ over the probability distribution together. The regularized objective can be written as follows:

$$\min_\rho \mathcal{F}[\rho] = F[\rho] + \lambda \cdot \mathrm{Ent}[\rho], \qquad F[\rho] = -J[\rho] + \frac{\lambda}{2} \cdot \mathbb{E}_\rho[\|\theta\|_2^2] + Z,$$

where $\lambda > 0$ is a regularization parameter and $Z > 0$ is a constant. Here we add two regularization terms to the objective function. The $L^2$-regularization $\mathbb{E}_\rho[\|\theta\|_2^2]$ helps to induce log-Sobolev inequality. This is due to the fact that $\|\theta\|_2^2$ is strongly convex, see Section B.1 especially Proposal 2 for details over log-Sobolev inequality. The entropy regularization term is required by adding Gaussian noise to the gradient, allowing global convergence analysis under less restrictive settings (Mei et al., 2019b). Adding these terms introduces a slight optimization bias of order $\mathcal{O}(\lambda)$. These regularization terms also have statistical benefits to smooth the problem. Note that we can rewrite the objective functional $\mathcal{F}$ as $\min_\rho \mathcal{F}[\rho] = -J[\rho] + \lambda \cdot D_{\mathrm{KL}}(\rho\|\nu)$ where $\nu = \mathcal{N}(0, I_d)$ is a standard Gaussian distribution.

In the sequel, we introduce the policy gradient with respect to the distribution $\rho$, the parameter of $\pi_\rho$.
**Proposition 1** (Policy Gradient). *For $\pi_\rho$ defined in Eq. (3), we have*

$$\frac{\delta J}{\delta \rho}[\rho](\theta) = -\mathbb{E}_{\sigma_{\pi_\rho}}[A_{\pi_\rho} \cdot h_\theta], \quad (4)$$

*where $\frac{\delta J}{\delta \rho}[\rho](\theta)$ is the first-variation of $J[\rho] = J[\pi_\rho]$ in Definition 1, and $A_{\pi_\rho}$ is the advantage function defined by $A_{\pi_\rho}(s,a) = Q_{\pi_\rho}(s,a) - \int \pi(\mathrm{d}a'|s) \cdot Q_{\pi_\rho}(s,a')$.*

See Appendix D.1 for the proof. Now we have all the elements necessary to construct the MFLPG. To obtain the optimal distribution $\rho^*$ that minimizes $\mathcal{F}[\rho]$, we define the surrogate first-variation of $F$, $\frac{\delta F}{\delta \rho}[\rho](\theta) = \mathbb{E}_{\sigma_{\pi_\rho}}[A_{\pi_\rho} \cdot h_\theta] + \frac{\lambda}{2}\|\theta\|_2^2$ by $g_t[\rho]$. Let the initial distribution $\rho_0 = \mathcal{N}(0, I_d)$. Then we update $\rho_t$ according to the following McKean-Vlasov stochastic differential equation with time $t \in \mathbb{R}_{\geq 0}$:

$$\mathrm{d}\theta_t = -\nabla g_t[\rho_t](\theta_t) \cdot \mathrm{d}t + \sqrt{2\lambda} \cdot \mathrm{d}W_t, \qquad g_t[\rho] = \mathbb{E}_{\sigma_{\pi_\rho}}[A_t \cdot h_\theta] + \frac{\lambda}{2}\|\theta\|_2^2, \qquad (5)$$

where $\theta_t \sim \rho_t(\mathrm{d}\theta)$, $A_t(s, a) = Q_t(s, a) - \int \pi_t(\mathrm{d}a'|s) \cdot Q_t(s, a')$ induced by the estimator $Q_t$ of $Q_{\pi_t}$ given by the critic, and $\{W_t\}_{t \geq 0}$ is the Brownian motion in $\mathbb{R}^d$ with $W_0 = 0$. It holds that the distribution of $\theta_t$ following the dynamics Eq. (5) solves the following Fokker-Planck equation:

$$\partial_t \rho_t = \lambda \cdot \Delta \rho_t + \nabla \cdot (\rho_t \cdot \nabla g_t[\rho_t]), \qquad (6)$$

Moreover, to utilize the nature of Wasserstein gradient flow, we denote by $\tilde{\rho}_t$ the approximated proximal Gibbs distribution (PGD) defined in Definition 2 around $\rho_t$, which is induced by $g_t[\rho_t]$ as $\tilde{\rho}_t = \exp\left(-\frac{1}{\lambda}g_t[\rho_t] - \ln \int \exp\left(-\frac{1}{\lambda}g_t[\rho_t]\right)\mathrm{d}\theta\right)$. If the exact value of the advantage function $A_{\pi_t}$ is available and $A_t = A_{\pi_t}$, then $\tilde{\rho}_t$ is propotional to $\exp(-\frac{1}{\lambda}\frac{\delta F}{\delta \rho}[\rho_t])$. In this point, the MFLD can evolve as $\rho_t$ locally approaches the PGD of $F$ around $\rho_t$. Indeed, Eq. (6) can be rewritten as a continuity equation like $\partial_t \rho_t = -\lambda \nabla \cdot (\rho_t \cdot v_t)$ with the velocity vector $v_t = -\nabla \ln \frac{\rho_t}{\tilde{\rho}_t}$.

**Discrete-time Analysis of MFLD** To implement our approach, we represent $\rho$ as a mixture of $m$ particles denoted by $\{\theta^{(i)}\}_{i=1}^m$, which corresponds to a neural network with $m$ neurons. We perform a discrete-time update at each $k$-th step of a noisy policy gradient method, where the policy parameter $\Theta = \{\theta^{(i)}\}_{i=1}^m$ is updated as in Algorithm 1. Note that, for each $k$-step, the agent uniformly sample $l \in [1, T_{TD}]$ and adopt $Q^{(l)}$ as $Q_k$ from the estimated Q-functions $\{Q^{(l)}\}_{l \in [T_{TD}]}$ obtained by MFLTD (Algorithm 2). Let $\eta > 0$ be a learning rate, and $K$ is the number of iterations. The discrete version of the MFLPG can be attributed to the MFLDs in Eq. (5) by taking the mean-field limit $m, k \to \infty$, $\eta \to 0$ and defining $t = \eta \cdot k$ and $T = \eta \cdot K$.

---

**Algorithm 1** Mean-field Langevin Policy Gradient

---

**Initialization:** $\theta_0^{(i)} \leftarrow N(0, I_d)$ for all $i \in [1, m]$ and $\pi_0(\cdot) \leftarrow \pi(\cdot; \{\theta_0\}_{i \in [1, m]})$.
1: **for** $k = 0$ to $K - 1$ **do**
2:      Given the current policy $\pi_k$, run Algorithm 2 and uniformly sample $l \in [T_{TD}]$: $Q_k \leftarrow Q^{(l)}$
3:      Calculate $A_k = Q_k - \langle \pi_k, Q_k \rangle$ and update $\Theta_{k+1} = \{\theta_{k+1}\}_{i \in [1, m]}$ with the Gaussian noise
     $\{\xi_k^{(i)}\}_{i \in [0, m]} \sim \mathcal{N}(0, I_d)$ by
     $\theta_{k+1}^{(i)} \leftarrow (1 - \eta \cdot \lambda) \cdot \theta_k^{(i)} - \eta \cdot \mathbb{E}_{\sigma_{\pi_k}}[A_k \cdot \nabla h_{\theta_k^{(i)}}] + \sqrt{2\lambda \cdot \eta} \cdot \xi_k^{(i)}$
4:      $\pi_{k+1}(\cdot) \leftarrow \pi(\cdot; \Theta_{k+1})$
5: **end for**
6: **return** $\pi_K$

---

## 3.2 CRITIC UPDATE: MEAN-FIELD LANGEVIN TD LEARNING

We now propose the MFLTD to approximately solve the problem (1) by optimizing a two-layer neural network in the mean field regime, and provide the algorithmic intuition. The difficulty in TD learning is that the semi-gradient of the mean-square Bellman error in (1) does not always point in the descent direction and it possibly does not converge. It is notable that it essentially stems from a nature of the mean-field regime such that it optimizes the probability measure instead of the parameter itself, that is, the optimization is performed as a Wasserstein gradient flow on the space of probability measure instead of that on an $L_2$ vector space like in the usual Euclidean space. Due to this fact, the semi-gradient does not provide a monotonic decrease of the objective in the mean-field regime while the normal gradient flow on a vector space decreases the objective monotonically. To resolve such a technical challenge, we propose a unique novel double-loop algorithm, MFLTD, like proximal gradient descent to make the algorithm monotonically decrease at each outer loop. MFLTD behaves like a majorization-minimization algorithm, where the inner loop solves the majorization problem and estimates the true value function from the fact that its minimum value always upper bounds the mean squared error, which is the true objective function.

**Outer loop** In the outer loop, the last iterate $Q^{(l)}$ of the previous inner loop is given. At the $l$-th step, the ideal minimizer of $\mathcal{L}_l$ given by the inner-loop MFLD lets the mean squared error be guaranteed to be upper bounded by the mean squared error at the previous step with KL-divergence regularization.

**Inner loop**    The inner loop is based on the KL-divergence regularized MFLD analysis in (Nitanda et al., 2022; Chizat, 2022). In the mean-field view, we minimize the objective $\min_q \mathcal{L}_l[q] = L_l[q] + \lambda_{\mathrm{TD}} \cdot \mathrm{Ent}[q]$ where $\lambda_{\mathrm{TD}}$ is a regularization parameter and $L_l[q]$ is defined, for $l \in [0, T_{\mathrm{TD}}]$, by

$$L_l[q] = \mathbb{E}_{\varsigma_\pi}[(Q^{(l)} - \mathcal{T}^\pi Q^{(l)}) \cdot (Q_q - Q_\pi)] + \frac{1}{2(1-\gamma)} \mathbb{E}_{\varsigma_\pi}[(Q^{(l)} - Q_q)^2] + \frac{\lambda_{\mathrm{TD}}}{2} \mathbb{E}_q[\|\omega\|_2^2] + Z, \quad (7)$$

where $Z > 0$ is a constant, on the right-hand side, the first term is the linearized surrogate TD error at $s$-th outer-loop step, the second one is the proximal regularization, and the last one is the $L^2$-regularization. We obtain the MFLD and the following Fokker-Planck equation at time $s$, respectively, as

$$\mathrm{d}\omega_s = -\nabla \frac{\delta L_l}{\delta q}[q_s](\omega_s) \cdot \mathrm{d}t + \sqrt{2\lambda_{\mathrm{TD}}} \cdot \mathrm{d}W_s, \quad \partial_s q_s = \lambda_{\mathrm{TD}} \cdot \Delta q_s + \nabla \cdot \left(q_s \nabla \frac{\delta L_l}{\delta q}[q_s]\right),$$

where $\{W_s\}_{s \geq 0}$ is the Brownian motion in $\mathbb{R}^d$ with $W_0 = 0$.

Let $(s', a')$ be the next state and action of $(s, a)$. To understand the intuition behind the proximal semi-gradient, Note that we have the gradient of first variation of $L_l$ as

$$\nabla \frac{\delta L_l}{\delta q}[q_s](\omega) = \mathbb{E}_{\varsigma_\pi}\left[\left(\bar{Q}_s^{(l)} - \mathcal{T}^\pi Q^{(l)}\right) \cdot \nabla h_\omega\right] + \lambda_{\mathrm{TD}} \cdot \omega,$$

where the expectation is obtained under $(s, a, s', a') \sim \varsigma_\pi$ and we define the averaged Q-function by $\bar{Q}_s^{(l)} = (Q_{q_s} - \gamma \cdot Q^{(l)})/(1-\gamma)$. See the corresponding part of Algorithm 2 for the discretization algorithm for inner-loop MFLDs.

We remark that considering that the inner-loop algorithm converges to the optimum at the exponential rate, the computational complexity of the inner-loop does not become a bottleneck in implementation. In this regard, the results in Section 5 offer valuable insights.

---

**Algorithm 2** Mean-field Langevin TD Learning

---

**Initialization:** $\tilde{\omega}_0^{(j)} \leftarrow N(0, I_d)$ for all $j \in [0, m_{TD}]$.
1: **for** $l = 0$ to $T_{\mathrm{TD}} - 1$ **do**
2:     **for** $r = 0$ to $K - 1$ **do**
3:         Average Q-function: $\bar{Q}_r^{(l)} = \frac{1}{1-\gamma}(Q_{\tilde{\Omega}_r} - \gamma \cdot Q^{(l)})$
4:         Run a noisy gradient descent for all $j \in [1, m]$:
           $\nabla \frac{\delta L_l}{\delta q}(\tilde{\omega}_r^{(j)}) \leftarrow \mathbb{E}_{\varsigma_\pi}\left[\left(\bar{Q}_r^{(l)}(x) - r(x) - \gamma \cdot Q^{(l)}(x')\right) \cdot \nabla h_{\tilde{\omega}_r^{(j)}}(x)\right] + \lambda_{\mathrm{TD}} \cdot \tilde{\omega}_r^{(j)}$
           $\tilde{\omega}_{r+1}^{(j)} \leftarrow \tilde{\omega}_r^{(j)} - \eta_{\mathrm{TD}} \cdot \nabla \frac{\delta L_l}{\delta q}(\tilde{\omega}_r^{(j)}) + \sqrt{2\lambda_{\mathrm{TD}}\eta_{\mathrm{TD}}} \cdot \xi_r^{(j)}, \quad \{\xi_r^{(j)}\}_{j \in [1,m]} \sim \mathcal{N}(0, I_d)$
5:     **end for**
6:     $Q^{(l)} \leftarrow Q(\cdot; \tilde{\Omega}^{(K)})$
7: **end for**
8: **return** $\{Q^{(l)}\}_{l \in [1, T_{\mathrm{TD}}]}$

---

## 4  MAIN RESULTS

In this section, we present the results of our investigation into the theoretical support of the mean-field Langevin actor-critic consisting of Algorithm 1 and 2. First of all, we base our analysis on the regularity condition that the reward is bounded.

**Assumption 2** (Regularity Condition on Reward). *We assume that there exists an absolute constant $R_r > 0$ such that $R_r = \sup_{(s,a) \in \mathcal{S} \times \mathcal{A}} |r(s, a)|$. As a result, we have $|V_\pi(s)| \leq R_r, |Q_\pi(s, a)| \leq R_r, |J[\pi]| \leq R_r$ and $|A_\pi(s, a)| \leq 2R_r$ for all $\pi$ and $(s, a) \in \mathcal{S} \times \mathcal{A}$.*

Considering Assumption 1 and 2, it holds that $R_r \leq R$ by setting $R > 0$ large enough where $R$ is the boundary of neural networks Q-function estimator. Such a regularity condition is commonly used in the literature (Liu et al., 2019; Wang et al., 2020). In what follows, we introduce the following regularity condition on the state-action value function $Q_\pi$.

**Assumption 3** (State-Action Value Function Class). *We define for $R, M > 0$*

$$\mathcal{F}_{R,M} = \left\{\int \beta' \cdot \sigma(w^\top(s, a, 1)) \cdot \rho'(\mathrm{d}\beta', \mathrm{d}w) : D_{\mathrm{KL}}(\rho\|\nu) \leq M, \ \rho' \in \mathscr{P}((-R, R) \times \mathbb{R}^{d-1})\right\},$$
$$(8)$$

*which is equivalent to the function class of $\mathbb{E}_{\theta \sim \rho}[h_\theta]$ for $\rho \in \mathscr{P}(\mathbb{R}^d)$. We assume that $Q_\pi(s,a) \in \mathcal{F}_{R,M}$ for any $\pi$.*

As will be further explained in Appendix B.2, we note that Assumption 3 is a natural regularity condition on $Q_\pi$, as $\mathcal{F}_{R,M}$ captures a rich family of functions, which is a subset of the Barron class (Barron, 1993). Indeed, by making the neural network radius $R, M$ sufficiently large, $\mathcal{F}_{R,M}$ asymptotically approaches the Barron class and captures a rich function class by the universal approximation theorem (Barron, 1993; Pinkus, 1999). Also, as long as smoothness and boundedness of networks are assumed (Assumption 1), every network can be included in the above class at least with a small modification. Similar regularity condition is a commonly used concept in literature (Farahmand et al., 2016; Yang & Wang, 2019; Liu et al., 2019; Wang et al., 2020).

**Error from Mean-field Langevin TD Learning**    In the continuous-time limit, we denote $q^{(l+1)}$ as the last-iterate distribution of the previous inner loop for each outer-loop step $l$, i.e., $q^{(l+1)} = q_S$ where $S$ is the inner-loop run-time. Regarding the outer-loop update, we obtain the following one-step descent lemma.

**Lemma 1** (One-Step Descent Lemma for MFLTD). *Let $q_*^{(l+1)}$ be the inner-loop optimal distribution for any inner step $l$. For $\{Q^{(l)}\}_{l \in [1, T_{\mathrm{TD}}]}$ in Algorithm 2 with the TD update in Line 6 and any $\gamma \in [0,1)$, it holds that*

$$\frac{\gamma(2-\gamma)}{2(1-\gamma)}\mathbb{E}_{\varsigma_\pi}\left[(\Delta Q^{(l+1)})^2 - (\Delta Q^{(l)})^2\right] \leq -\frac{1-\gamma}{2}\mathbb{E}_{\varsigma_\pi}[(\Delta Q^{(l+1)})^2] + \frac{2R}{1-\gamma}(\mathbb{E}_{\varsigma_\pi}[(Q^{(l+1)} - Q_*^{(l+1)})^2])^{\frac{1}{2}}$$
$$+ \lambda_{\mathrm{TD}} \cdot D_{\mathrm{KL}}(q^{(l+1)}\|q_*^{(l+1)}) + \lambda_{\mathrm{TD}} \cdot D_{\mathrm{KL}}(q_\pi\|\nu),$$
(9)

*where we define that $\Delta Q^{(l)} = Q^{(l)} - Q_\pi$, and denote $Q_*^{(l+1)}$ as a Q-function $Q_{q_*^{(l+1)}}$.*

See Appendix C.1 Lemma 1 shows that the Q-function of the outer steps of the MFLTD $Q^{(l)}$ converges to the true state-value function $Q_\pi$. The second and third term of the right-hand side of Eq. (9) represents non-asymptotic errors obtained through the inner loop, and it exponentially decreases with an increase in the run-time $S$ of the inner loop. The key to the proof of Lemma 1 is the use of geometric features due to the fact that the norm of the Bellman equation operator is no more than 1 (Lemma 7). The shrinking norm suppresses errors in the semi-gradient direction that deviates from the true gradient direction. Combining Proposition 5, in what follows, Lemma 1 allows us to establish the global convergence theorem for the MFLTD as

**Theorem 1** (Global Convergence of the MFLTD). *Under Assumption 1, 2, and 3, the outputs $\{Q^{(l)}\}_{l=1}^{T_{\mathrm{TD}}}$ of Algorithm 2 satisfies, for the inner run time $S > 0$, that*

$$\frac{1}{T_{\mathrm{TD}}}\sum_{l=1}^{T_{\mathrm{TD}}}\mathbb{E}_{\varsigma_\pi}[(Q^{(l)} - Q_\pi)^2] \leq \frac{4\gamma(2-\gamma)R^2}{(1-\gamma)^2 T_{\mathrm{TD}}} + C_1 e^{(-\alpha\lambda_{\mathrm{TD}}S)} + C_2\lambda_{\mathrm{TD}}e^{(-2\alpha\lambda_{\mathrm{TD}}S)} + C_3\lambda_{\mathrm{TD}},$$

*where we denote by $C_1, C_2, C_3 > 0$ the absolute constants satisfying that $C_1 = \frac{8(3-2\gamma)^{\frac{1}{2}}R^3}{(1-\gamma)^{\frac{3}{2}}}, C_2 = \frac{8(3-2\gamma)R^4}{(1-\gamma)^2}, C_3 = \frac{2M}{1-\gamma}$, and we define $\alpha$ as a LSI constant defined in Definition 2.*

See Appendix C.2 for the proof sketch. Theorem 1 shows that, given a policy $\pi$, the MFLTD converges Q-function to the true state-action value function $Q_\pi$ at the time-averaged sublinear rate $\mathcal{O}(1/T_{\mathrm{TD}})$ for the iteration number $T_{\mathrm{TD}}$ of the outer loop. This result is in perfect agreement with the convergence rate $\mathcal{O}(1/T_{\mathrm{TD}})$ that Cai et al. (2019) obtains from TD learning in the NTK regime and the convergence rate $\mathcal{O}(1/T_{\mathrm{TD}})$ that Zhang et al. (2020) obtains from TD learning viewed as the Wasserstein gradient flow attributed to lazy training. Note here that the results obtained in this study ignore the computational speed of the inner loop, which converges at an exponential rate. However, it is worth noting that this is the first time that global convergence has been demonstrated in a domain that takes advantage of the data-dependent advantage of neural networks. Since the bias $\mathcal{O}(\lambda_{\mathrm{TD}})$ in this result is due only to the inner-loop algorithm, we follow Theorem 4.1 in (Chizat, 2022) and can achieve the annealed Langevin dynamics by attenuating $\lambda_{\mathrm{TD}}$ by $\mathcal{O}(1/\log(S))$.

**Global Convergence of the MFLPG**    We lay out the analysis of convergence and global optimality of the MFLPG in Algorithm 1. In our algorithm, since MFLD can be attributed to the Wasserstein gradient flow, the convergence to the stationary point is guaranteed.

**Lemma 2** (Time Derivative of Objective Function). *Under Assumption 1, 2, and 3, for any $\tilde{\rho} \in \mathcal{P}_2, \beta > 0$, we obtain that*

$$\frac{\mathrm{d}}{\mathrm{d}t}\mathcal{F}[\rho_t] \leq -\alpha\lambda \cdot \mathbb{E}_{\sigma_t}\left[A_{\pi_t} \cdot \left(f_t - \beta \cdot \tilde{f}\right)\right] + \alpha\lambda^2 \cdot (\beta \cdot D_{\mathrm{KL}}\left(\tilde{\rho}\|\nu\right) - D_{\mathrm{KL}}\left(\rho_t\|\nu\right)) + 2L_2^2 \cdot \Delta_t, \quad (10)$$

*where $\tilde{f} = \int h_\theta \tilde{\rho}(\mathrm{d}\theta)$, $\alpha > 0$ is the LSI constant of $\hat{\rho}_t$ and $\Delta_t = \|\mathrm{d}\sigma_t/\mathrm{d}\varsigma_t\|_{\varsigma_t,2}^2 \cdot \mathbb{E}_{\sigma_t}[(Q_t - Q_{\pi_t})^2]$ is the critic error where $Q_t$ is the Q-function estimator given by the critic.*

See Appendix D.2 for the proof. The first term on the right-hand side of Eq. (10) is derived from the equivalent of the square of the gradient of the expected total rewards $J[\rho_t]$. It is worth noting that if $J$ is a convex function, we can substitute $\rho^*$ for this arbitrary $\tilde{\rho}$ and this part appears as the performance difference. Meanwhile, the second term is the regularization error, and the third term is the policy evaluation error given by the critic. We, therefore, wish to suppress this first term from the equality obtained by Proposition 3 which establishes the one-point convexity of $J[\pi]$ at the global optimum $\pi^*$ derived by Kakade & Langford (2002). In what follows, we lay out a moment condition on the discrepancy between the state-action visitation measure $\sigma_t$ and the stationary state-action distribution $\varsigma_t$ corresponding to the same policy $\pi_t$, and also optimal policy $\pi^*$.

**Assumption 4** (Moment Condition on Radon-Nikodym Derivative). *We assume that there exists absolute constants $\kappa, \iota > 0$ such that for all $t \in [0, T]$*

$$\|\mathrm{d}\sigma_t/\mathrm{d}\varsigma_t\|_{\varsigma_t,2}^2 \leq \iota, \qquad \|\mathrm{d}\sigma^*/\mathrm{d}\sigma_t\|_{\sigma_t,2}^2 \leq \kappa,$$

*where $\frac{\mathrm{d}\sigma_t}{\mathrm{d}\varsigma_t}$ and $\frac{\mathrm{d}\sigma^*}{\mathrm{d}\sigma_t}$ are the Radon-Nikodym derivatives.*

It is important to note that when the MDP starts at the stationary distribution $\varsigma_t$, the state-action visitation measures $\sigma_t$ are identical to $\varsigma_t$. Additionally, if the induced Markov state-action chain rapidly reaches equilibrium, this assumption also holds true. The same requirement is imposed by Liu et al. (2019); Wang et al. (2020). Meanwhile, the optimal moment condition in Assumption 4 asserts that the concentrability coefficients are upper-bounded. This regularity condition is a commonly used concept in literature (Farahmand et al., 2016; Chen & Jiang, 2019; Liu et al., 2019; Wang et al., 2020). Finally, we lay out the following regularity condition on the richness of the function class as

**Assumption 5** (Regularity Condition on $\mathcal{F}_{R,M}$). *We assume that there exists a constant $M, B > 0$ such that there exists a function $f \in \mathcal{F}_{R,M}$ satisfying that $|\langle A_\pi, f\rangle_{\sigma_\pi}|/\|A_\pi\|_{\sigma_\pi} \geq 1/B$ for each $\pi$.*

Assumption 5 guarantees that when one has a policy, one can always approximate the advantage function in the gradient direction of the policy gradient within the finite KL-divergence ball. Indeed, for example, Assumption 5 is satisfied when $A_\pi/\|A_\pi\|_{\sigma_\pi} \in \mathcal{F}_{R,M}$. Now that $Q_\pi \in \mathcal{F}_{R,M}$ is assumed by Assumption 3, the geometric regularity of Assumption 5, coupled with the richness of the function class $\mathcal{F}_{R,M}$, is moderate. See Appendix B.2 for details. In what follows, we establish the global optimality and the convergence rate of the MFLPG.

**Theorem 2** (Global Optimality and Convergence of the MFLPG). *Let $J^*$ be the optimal expected total reward. We set $\lambda_{\mathrm{TD}} = \alpha\lambda^2$ and $T_{\mathrm{TD}} = \mathcal{O}(1/\lambda_{\mathrm{TD}})$. Under the assumptions of Lemma 2 and Assumption 4, 5, by Algorithm 1, where the actor update is given in Eq. (5), we obtain for all $T \in \mathbb{R}$ and $\lambda > 0$ that*

$$J^* - J[\rho_T] \leq \exp(-2\alpha\lambda T) \cdot (J^* - J[\rho_0]) + \mathcal{O}\left(\lambda\right). \quad (11)$$

*Proof.* We utilize a one-point convexity of the expected total rewards in Kakade & Langford (2002) to prove the global optimality of the stationary point led by Lemma 2. We here use the geometric property of the richness of the approximation capacity of $\mathcal{F}_{R,M}$ to connect this one-point convexity. See Appendix D.3 for a detailed proof. $\qquad\square$

Theorem 2 shows that the suboptimality of the sequence of actors returned by MFLPG converges linearly to zero up to a $\mathcal{O}(\lambda)$ bias induced by the KL-divergence regularization. Here the suboptimality is in terms of the *unregularized expected total reward $J$* and $\lambda$ can be as close to 0. Therefore, by choosing a sufficiently small $\lambda$, we conclude that MFLPG finds the globally optimal policy efficiently. In addition, as in other general regularized optimization algorithms, there is always a trade-off between the convergence rate in the first term and the bias term in the second term, on the right-hand side of Eq. (11). In comparison, by redefining the regularization coefficient $\lambda$ as a time-dependent variable $\lambda_t$ by $\lambda_t = \mathcal{O}(1/\ln t)$, Chizat (2022) established the objective difference converges to the globally optimal objective at a sublinear convergence rate with no error term in general MFLD problems. Therefore, we highlight that Theorem 4.1 in Chizat (2022) also guarantees the sublinear

convergence of Theorem 2 without regularization bias. To the best of our knowledge, this is the first analysis that shows the linear convergence $\mathcal{O}(\exp(-\alpha\lambda T))$ to globally optimal expected total rewards $J^*$ in neural policy gradient and neural actor-critic. This predominates the conventional convergence rate $\mathcal{O}(T^{-1/2})$ in neural policy gradients with the NTK-type analysis (Wang et al., 2020). This is also the first convergence analysis of algorithms trained as neural networks for both actors and critics, where the feature representation (Suzuki, 2019; Ghorbani et al., 2019) is guaranteed to be learned in a data-dependent manner beyond the lazy-training regime.

## 5 NUMERICAL ANALYSIS

In this section, we conducted a numerical experiment to compare the Critic component, which is based on the proposed MFLTD, against the existing TD(1) algorithm that utilizes the Bellman error semi-gradient. Additionally, we demonstrated how the learning performance differs when using a neural network that follows the NTK with a representation that is independent of input data and dependent on initial values. Specifically, we performed learning on the CartPole-v1 environment provided by OpenAI's Gym and implemented the estimation of the state-action value function during optimal policy selection. In this experiment, we used a neural network with 256 neurons, ran 4000 episodes, and employed a learning rate of $\eta = 0.0001$ for MFLTD. Notably, we conducted MFLTD's inner loop with a step size of $K = 10$, repeated it $T_{\text{TD}} = 400$ times in the outer loop, and sampled using one episode for each inner step. Furthermore, we applied Gaussian noise of magnitude induced by the entropy regularization parameter $\lambda = 0.001$, following Algorithm 2, along with $L_2$ regularization. To assess the difference in performance due to representation learning covered by Mean-field analysis, we also implemented NTK-TD with a double-loop setup where representations are fixed at initial values, similar to MFLTD. Additionally, we addressed the primary weakness of our proposed algorithm, the double-loop, and examined its impact on computational complexity. To do so, we ran the conventional single-loop TD(1) algorithm under the same conditions.

Figure 1 presents the average and standard deviation of each learning process conducted ten times. It's essential to note that Figure 1 compares the results under the same number of episodes and parameter updates. From this figure, we observe that learning with features independent of initial values outperforms when compared with an equal number of neurons, primarily due to increased expressiveness gained through feature learning. Furthermore, while the single-loop results are faster in regions of lower accuracy under the same computational load and time, they exhibit decreased speed in regions of higher accuracy, ultimately demonstrating that our proposed double-loop method approximates the true value function more effectively.

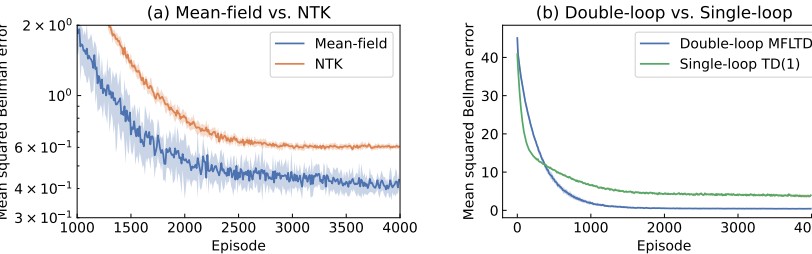

Figure 1: Comparison of Time Evolution of Mean Squared Bellman Error Between Algorithms for TD Learning Near the Optimal Policy in the Game Model "CartPole-v1".

## 6 CONCLUSION

We studied neural policy optimization in the mean-field regime, and provided the first global optimality guarantee and the linear convergence rate for a neural actor-critic algorithm, in the presence of feature learning. For both actor and critic, we attributed their updates to the mean-field Langevin dynamics and analyzed their evolutions as the optimization of corresponding probability measures. We provide theoretical guarantees for global convergence to global optimality, and empirical experiments that validate the superiority of the proposed algorithm in policy evaluation. In future work, it would be interesting to extend our analysis to the finite particles, discrete-time, and stochastic gradient settings Suzuki et al. (2023).

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

## TABLE OF CONTENTS

APPENDIX

# A    NOTATIONS

We denote by $\mathscr{P}(\mathscr{X})$ the set of distribution measures over the measurable space $\mathscr{X}$. Given a distribution measure function $\mu \in \mathscr{P}(\mathscr{X})$, the expectation with respect to $\mu$ as $\mathbb{E}_{\theta \sim \mu}[\cdot]$ or simply $\mathbb{E}_{\mu}[\cdot], \mathbb{E}_{\theta}[\cdot]$ when the random variable and distribution are obvious from the context. In addition, for $\mu \in \mathscr{P}(\mathscr{X})$ and $p > 0$, we define $\|f(\cdot)\|_{\mu,p} = (\int_{\Theta} |f|^p \mathrm{d}\mu)^{\frac{1}{p}}$ as the $L^p(\mu)$-norm of $f$. We define $\|f(\cdot)\|_{\mu,\infty} = \inf\{C \geq 0 : |f(x)| \leq C \text{ for } \mu\text{-almost every } x\}$ as the $L^{\infty}(\mu)$-norm of $f$. We write $\|f\|_{\mu,p}$ for notational simplicity when the variable of $f$ is obvious from the context. Especially, the $L_2(\mu)$-norm is denoted by $\|\cdot\|_{\mu}$. For a vector $v \in \mathbb{R}^d$ and $p > 0$, we denote by $\|v\|_p$ the $L^p$-norm of $v$. Given two distribution measures $\mu, \rho \in \mathscr{P}(\mathscr{X})$, we denote the Radon–Nikodým derivative between $\mu$ and $\rho$ by $\frac{\mathrm{d}\mu}{\mathrm{d}\rho}$. $D_{\mathrm{KL}}(\cdot\|\cdot)$ stands for the Kullbuck-Leibler divergence as $D_{\mathrm{KL}}(\mu\|\rho) = \int \mathrm{d}\mu \ln \frac{\mathrm{d}\mu}{\mathrm{d}\rho}$, and also $\mathrm{I}(\cdot\|\cdot)$ stands for the Fisher divergence as $\mathrm{I}(\mu\|\rho) = \int \mathrm{d}\mu \|\nabla_{\theta} \ln \frac{\mathrm{d}\mu}{\mathrm{d}\rho}\|_2^2$. Also, we define the entropy $\mathrm{Ent}[\cdot]$ by $\mathrm{Ent}[\mu] = \int \mathrm{d}\mu \ln \mu$. Let $\mathcal{P}_2 \subset \mathscr{P}(\mathbb{R}^d)$ be the space of probability density functions such that both the entropy and second moment are finite.

# B    ADDITIONAL REMARKS

## B.1    LOGARITHMIC SOBOLEV INEQUALITY

In this paper, we extend the convergence analysis of a nonlinear Fokker-Planck equation, mean-field Langevin dynamics to the context of reinforcement learning. The analysis is based on the KL-divergence regularization (Mei et al., 2019a; Hu et al., 2021; Chen et al., 2020) and the induced log-Sobolev inequality (Nitanda et al., 2022; Chizat, 2022). Below are some mathematical tools necessary for them. Particularly, in the MFLD convergence analysis, it is important to make use of the following proximal Gibbs distribution defined as follows. To define the MFLD of functional $F$, we first introduce the first variation of functionals as

**Definition 1** (First-variation of Functionals). Let $F : \mathcal{P}_2 \to \mathbb{R}$ and we suppose there is a functional $\frac{\delta F}{\delta \rho} : \mathcal{P}_2 \times \mathbb{R}^d \ni (\rho, \theta) \mapsto \frac{\delta F}{\delta \rho}[\rho](\theta) \in \mathbb{R}$ such that for any $\rho, \rho' \in \mathcal{P}_2$,

$$\frac{\mathrm{d}F(\rho + \epsilon \cdot (\rho' - \rho))}{\mathrm{d}\epsilon}\bigg|_{\epsilon=0} = \int \frac{\delta F}{\delta \rho}[\rho](\theta)(\rho' - \rho)(\mathrm{d}\theta),$$

for all $\rho \in \mathcal{P}_2$. If there exists a functional $\frac{\delta F}{\delta \rho}[\rho](\theta)$, we say that $F$ is differentiable at $\rho$.

Note that any first variation of a functional is invariant with respect to a constant shift. In what follows, we define the proximal Gibbs distribution (PGD) with a first variation, as

**Definition 2** (Proximal Gibbs Distribution (PGD)). Let $\rho \in \mathcal{P}_2$ and $\lambda > 0$ the temperature. We define the Gibbs distribution with potential function $-\frac{1}{\lambda}\frac{\delta F}{\delta \rho}$ around $\rho$ by

$$\hat{\rho}(\theta) \propto \exp\left(-\frac{1}{\lambda}\frac{\delta F}{\delta \rho}[\rho](\theta)\right),$$

where $\theta \in \mathbb{R}^d$. We call $\hat{\rho}(\theta)$ the proximal Gibbs distribution of the functional $F$ around $\rho$.

The convergence analysis of $\rho_t$ over the objective $F$ heavily depends on the relationship between the PGD around $\rho_t$, $\hat{\rho}_t$ and the optimal distribution $\rho^*$. Regarding the convergence rate of MFLDs, the key analysis is depending on the following logarithmic Sobolev inequality.

**Definition 3** (Logarithmic Sobolev Inequality (LSI)). We define that a distribution measure $\rho \in \mathcal{P}_2$ satisfies a logarithmic Sobolev inequality with constant $\alpha > 0$, which is called LSI($\alpha$) in short, if and only if, for any smooth function $\Psi : \mathbb{R}^d \to \mathbb{R}$ with $\mathbb{E}_{\rho}[\Psi^2] < \infty$, it holds that

$$\mathbb{E}_{\rho}[\Psi^2 \ln(\Psi^2)] - \mathbb{E}_{\rho}[\Psi^2] \cdot \ln(\mathbb{E}_{\rho}[\Psi^2]) \leq \frac{2}{\alpha}\mathbb{E}_{\rho}[\|\nabla \Psi\|_2^2],$$

which is equivalent to the condition that for all $\nu \in \mathcal{P}_2$ absolutely continuous w.r.t. $\mu$, it holds

$$D_{\mathrm{KL}}(\rho\|\mu) \leq \frac{1}{2\alpha}\mathrm{I}(\rho\|\mu).$$

In particular, the LSI holds uniformly for the PGD over the mean-field neural network condition, given some appropriate boundedness assumptions. The result is achieved by leveraging two well-known facts. Firstly, it is established that strongly log-concave densities satisfy the LSI with a dimension-free constant, up to the spectral norm of the covariance. For instance, Bakry & Émery (1985) showed the following lemma:

**Lemma 3** (Bakry & Émery (1985))**.** *If $\rho \propto \exp(-f(\theta))$ is a smooth probability density with $f : \mathbb{R}^d \to \mathbb{R}$ and there exists $c > 0$ such that the Hessian matrix of $f$ satisfies $\nabla^2 f \succeq c \cdot I_d$, then the distribution $\rho(\theta)\mathrm{d}\theta$ satisfies the LSI with constant $c$.*

It is worth noting that, for example, the Gaussian distribution $\nu \sim \mathcal{N}(0, I_d)$ satisfies Lemma 3 with the LSI constant $c = 1$. That is, $\nu \sim \mathcal{N}(0, I_d)$ satisfies LSI(1). In addition to that, preservation of LSI under bounded perturbation has been demonstrated in Holley & Stroock (1987) as

**Lemma 4** (Holley & Stroock (1987))**.** *If $\rho$ is a distribution on $\mathbb{R}^d$ that satisfies the LSI with constant $c > 0$, and for a bounded function $f : \mathbb{R}^d \to \mathbb{R}$, the distribution $\rho_f$ is defined as*

$$\rho_f(\theta) \propto \exp(f(\theta)) \cdot \rho(\theta),$$

*then $\rho_f$ satisfies the LSI with a constant $c/\exp\left(4|f|_\infty\right)$.*

Combined with the previous example of Lemma 3, $\nu_f$ with some uniformly bounded potential function $f$ satisfies Lemma 4. These lemmas lead to the important fact that follows. Under the definition of the two-layer neural network in mean-field regime and Assumption 1, the PGD of each function appearing in this paper satisfies the LSI with an absolute constant $\alpha$. Specifically, we have

**Proposition 2** (LSI Constant of PGD)**.** *Let the first-variation of a function $L$, $\frac{\delta L}{\delta \rho}$ be uniformly bound by $C > 0$, and $F = L + \frac{\lambda}{2} \cdot \mathbb{E}_\rho[\|\theta\|_2^2]$ with $\lambda > 0$. Then we have that the PGD around $\rho$, $\hat{\rho}$ satisfies the LSI with a constant $\alpha = \frac{1}{\exp(\frac{4C}{\lambda})}$.*

In our case, the boundness of each first-variation is guaranteed by the neural network's boundness in Assumption 1 and the reward's boundness in Assumption 2. It is worth noting that the exponential dependence on the LSI constant may be inevitable in the most general setting (Menz & Schlichting, 2014).

## B.2 ON THE FUNCTION CLASS

In Assumption 3 and 5, we considered the class of measures with the bounded KL divergence and some regularity condition. We first note that, as we let $M$ and $B$ large, then $\mathcal{F}_{R,M}$ satisfying Assumption 5 can contain a wider class of neural networks. What is worth mentioning is the relation to the so-called Barron class. As we increase $M$ and $B$, we can approximate a neural network in the Barron class with arbitrary accuracy.

Barron (1993; 1994) showed that a neural network with a sigmoid activation function can avoid the curse of dimensionality (Weinan et al., 2019) if the Fourier transform of the function $f$ satisfies certain integrability conditions, and he defined a function class with good properties that can be approximated universally (Barron, 1993; Pinkus, 1999). Particularly, we name the function class as the Barron class and denote it as $\mathcal{B}_{\mathcal{F}}$, such that

$$\int_{\mathbb{C}^d} \|\omega\|_1^2 \cdot |\hat{f}(\omega)| \mathrm{d}\omega < \infty,$$

where $\hat{f}$ is the Fourier transform of a function $f$.

One pleasant aspect of considering the Barron class is that one of the biggest contributions of feature learning, the avoidance of the curse of dimensionality inherent in neural networks, theoretically arises (Weinan et al., 2019). The Barron class is also closely related to the avoidance of the curse of dimensionality in other function spaces such as in the mixed Besov space (Suzuki, 2019).

A similar analysis of function classes has been developed (Klusowski & Barron, 2016; E et al., 2019) and, in particular, the following derivations of the Barron class are known:

**Definition 4** (Barron Class (Li et al., 2020))**.** The Barron class is defined as

$$\mathcal{B}_\infty = \left\{ \int_{\mathbb{R}^d} \beta(w) \cdot \sigma(w^\top(x, 1)) \cdot \rho(\mathrm{d}w) \; : \; \rho \in \mathscr{P}(\mathbb{R}^d), \; \inf_\rho \|\beta(w) \cdot (\|w\|_1 + 1)\|_{\rho,\infty} < \infty \right\},$$

The Barron norm for any $f \in \mathcal{B}_\infty$ is defined by $\|f\|_{\mathcal{B}_\infty} = \inf_\rho \|\beta(w) \cdot (\|w\|_1 + 1)\|_{\rho,\infty}$. In addition, we define the R-Barron space by $R' > 0$ by

$$\mathcal{B}_{R'} = \{f \in \mathcal{B}_\infty \ : \ \|f\|_{\mathcal{B}_\infty} \leq R'\}.$$

Note that the R-Barron space $\mathcal{B}_{R'}$ corresponds to the function class $\mathcal{F}_{R,M}$ targeted by our neural network. That is, our function class can approximate an element of the Barron class with any degree of accuracy as a set. Although the R-Barron space and the Barron class cannot be directly compared, they are closely related and can be adequately covered by a sufficiently large $R'$.

Finally, we remark that based on Assumption 1, which guarantees the smoothness and boundedness of neural networks, it is very easy for such a network to satisfy Eq. (8) with some $R$ and $K$ at least with a small modification. For any such neural network, if we consider convolution of the corresponding measure with a Gaussian of small variance, this does not change the output of the network very much due to the smoothness and boundedness, this smoothens the distribution and as a result, guarantees that the modified neural network belongs to our class of measures.

## C  MEAN-FIELD LANGEVIN TD LEARNING

### C.1  PROOF OF LEMMA 1

*Proof.* From the definition of $L_l[\cdot]$ in Eq. (7), for $s \in [0, T_{\mathrm{TD}}]$ we have

$$
\begin{aligned}
\mathcal{L}_l[q] =& L_l[q] + \lambda_{\mathrm{TD}} \cdot \mathrm{Ent}[q] \\
=& \mathbb{E}_{\varsigma_\pi}[(Q^{(l)} - \mathcal{T}Q^{(l)}) \cdot (Q_q - Q_\pi)] + \frac{1}{2(1-\gamma)} \mathbb{E}_{\varsigma_\pi}[(Q^{(l)} - Q_q)^2] + \lambda_{\mathrm{TD}} \cdot D_{\mathrm{KL}}(q\|\nu)
\end{aligned}
\tag{12}
$$

The inner algorithm performs a gradient descent of $\mathcal{L}_l$ over Wasserstein metric; note that $L_l$ always upper bounds the mean squared error. Therefore, we evaluate the difference of the objective function $L_l$ between the optimum of the true objective function, $q_\pi$, and $q^{(l+1)}$ which is ideally the optimum of the Majorization problem, from above and below, respectively. For $l \in \mathbb{N}$ we have

$$
\begin{aligned}
\mathcal{L}_l[q_\pi] - \mathcal{L}_l[q^{(l+1)}] =& -\mathbb{E}_{\varsigma_\pi}[(Q^{(l)} - \mathcal{T}Q^{(l)}) \cdot (Q^{(l+1)} - Q_\pi)] + \frac{1}{2(1-\gamma)} \mathbb{E}_{\varsigma_\pi}[(Q^{(l)} - Q_\pi)^2] \\
& - \frac{1}{2(1-\gamma)} \mathbb{E}_{\varsigma_\pi}[(Q^{(l)} - Q^{(l+1)})^2] + \lambda_{\mathrm{TD}} \cdot D_{\mathrm{KL}}(q_\pi\|\nu) - \lambda_{\mathrm{TD}} \cdot D_{\mathrm{KL}}(q^{(l+1)}\|\nu).
\end{aligned}
\tag{13}
$$

In what follows, we upper bound the first term on the right-hand side of Eq. (13) by each difference of Q-functions without any transition kernels. For simplicity, we define that $\Delta Q^{(l)} = Q^{(l)} - Q_\pi$, $I$ is an identity operator, and $\mathcal{P} : L^2(\varsigma_\pi)(\mathcal{S} \times \mathcal{A}) \to L^2(\varsigma_\pi)(\mathcal{S} \times \mathcal{A})$ as the linear operator such that $\mathcal{P}Q(s,a) = \int \mathrm{d}s' P(s'|s,a) \int \mathrm{d}a' \pi(a'|s') Q(s,a)$, $Q \in L^2(\varsigma_\pi)(\mathcal{S} \times \mathcal{A})$. Focusing on the fact that we can reformulate the first term on the right-hand side of Eq. (13) as $\mathbb{E}_{\varsigma_\pi}[(Q^{(l)} - \mathcal{T}Q^{(l)}) \cdot (Q^{(l+1)} - Q_\pi)] = \mathbb{E}_{\varsigma_\pi}[\Delta Q^{(l+1)}(I - \gamma\mathcal{P})\Delta Q^{(l)}]$, it holds that

$$
\begin{aligned}
& \mathbb{E}_{\varsigma_\pi}[(\Delta Q^{(l+1)} - \Delta Q^{(l)})(I - \gamma\mathcal{P})(\Delta Q^{(l+1)} - \Delta Q^{(l)})] \\
=& \mathbb{E}_{\varsigma_\pi}[\Delta Q^{(l+1)}(I - \gamma\mathcal{P})\Delta Q^{(l+1)}] + \mathbb{E}_{\varsigma_\pi}[\Delta Q^{(l)}(I - \gamma\mathcal{P})\Delta Q^{(l)}] \\
& - \mathbb{E}_{\varsigma_\pi}[\Delta Q^{(l+1)}(I - \gamma\mathcal{P})\Delta Q^{(l)}] - \mathbb{E}_{\varsigma_\pi}[\Delta Q^{(l)}(I - \gamma\mathcal{P})\Delta Q^{(l+1)}] \\
=& \mathbb{E}_{\varsigma_\pi}[\Delta Q^{(l+1)}(I - \gamma\mathcal{P})\Delta Q^{(l+1)}] + \mathbb{E}_{\varsigma_\pi}[\Delta Q^{(l)}(I - \gamma\mathcal{P})\Delta Q^{(l)}] \\
& + \mathbb{E}_{\varsigma_\pi}[\Delta Q^{(l)}(I - \gamma\mathcal{P}^*)\Delta Q^{(l+1)}] - \mathbb{E}_{\varsigma_\pi}[\Delta Q^{(l)}(I - \gamma\mathcal{P})\Delta Q^{(l+1)}] \\
& - 2\mathbb{E}_{\varsigma_\pi}[\Delta Q^{(l+1)}(I - \gamma\mathcal{P})\Delta Q^{(l)}] \\
=& \mathbb{E}_{\varsigma_\pi}[\Delta Q^{(l+1)}(I - \gamma\mathcal{P})\Delta Q^{(l+1)}] + \mathbb{E}_{\varsigma_\pi}[\Delta Q^{(l)}(I - \gamma\mathcal{P})\Delta Q^{(l)}] \\
& + \gamma \cdot \mathbb{E}_{\varsigma_\pi}[\Delta Q^{(l+1)}(\mathcal{P}^* - \mathcal{P})\Delta Q^{(l)}] - 2\mathbb{E}_{\varsigma_\pi}[\Delta Q^{(l+1)}(I - \gamma\mathcal{P})\Delta Q^{(l)}]
\end{aligned}
\tag{14}
$$

where $\mathcal{P}^*$ is the adjoint operator of $\mathcal{P}$. As for each term of Eq. (14), we have the following inequalities:

$$
\begin{aligned}
&\mathbb{E}_{\varsigma_\pi}[(\Delta Q^{(l+1)} - \Delta Q^{(l)})(I - \gamma \mathcal{P})(\Delta Q^{(l+1)} - \Delta Q^{(l)})] \\
&= \mathbb{E}_{\varsigma_\pi}[(Q^{(l+1)} - Q^{(l)})(I - \gamma \mathcal{P})(Q^{(l+1)} - Q^{(l)})] \\
&= \mathbb{E}_{\varsigma_\pi}[(Q^{(l+1)} - Q^{(l)})^2] - \gamma \cdot \mathbb{E}_{\varsigma_\pi}[(Q^{(l+1)} - Q^{(l)}) \cdot \mathcal{P}(Q^{(l+1)} - Q^{(l)})] \\
&\leq \mathbb{E}_{\varsigma_\pi}[(Q^{(l+1)} - Q^{(l)})^2] + \gamma \cdot \mathbb{E}_{\varsigma_\pi}[(Q^{(l+1)} - Q^{(l)})^2]^{1/2} \cdot \|\mathcal{P}(Q^{(l+1)} - Q^{(l)})\|_{\varsigma_\pi,2} \\
&\leq \mathbb{E}_{\varsigma_\pi}[(Q^{(l+1)} - Q^{(l)})^2] + \gamma \cdot \mathbb{E}_{\varsigma_\pi}[(Q^{(l+1)} - Q^{(l)})^2]^{1/2} \cdot \mathbb{E}_{\varsigma_\pi}[(Q^{(l+1)} - Q^{(l)})^2]^{1/2} \\
&= (1 + \gamma) \cdot \mathbb{E}_{\varsigma_\pi}[(Q^{(l+1)} - Q^{(l)})^2],
\end{aligned}
\tag{15}
$$

where the first inequality follows from Hölder's inequality and the second one follows from Lemma 7. In exactly the same way, we have

$$
\begin{aligned}
&-\mathbb{E}_{\varsigma_\pi}[\Delta Q^{(l+1)}(I - \gamma \mathcal{P})\Delta Q^{(l+1)}] \\
&= -\mathbb{E}_{\varsigma_\pi}[(\Delta Q^{(l+1)})^2] + \gamma \cdot \mathbb{E}_{\varsigma_\pi}[\Delta Q^{(l+1)} \cdot \mathcal{P}\Delta Q^{(l+1)}] \\
&\leq -\mathbb{E}_{\varsigma_\pi}[(\Delta Q^{(l+1)})^2] + \gamma \cdot \mathbb{E}_{\varsigma_\pi}[(\Delta Q^{(l+1)})^2]^{1/2} \cdot \|\mathcal{P}(\Delta Q^{(l+1)})\|_{\varsigma_\pi,2} \\
&\leq -\mathbb{E}_{\varsigma_\pi}[(\Delta Q^{(l+1)})^2] + \gamma \cdot \mathbb{E}_{\varsigma_\pi}[(\Delta Q^{(l+1)})^2]^{1/2} \cdot \mathbb{E}_{\varsigma_\pi}[(\Delta Q^{(l+1)})^2]^{1/2} \\
&= -(1 - \gamma) \cdot \mathbb{E}_{\varsigma_\pi}[(\Delta Q^{(l+1)})^2],
\end{aligned}
\tag{16}
$$

where the first inequality follows from Hölder's inequality and the second one follows from Lemma 7. From the same discussions, we also have $-\mathbb{E}_{\varsigma_\pi}[\Delta Q^{(l)}(I - \gamma \mathcal{P})\Delta Q^{(l)}] \leq -(1 - \gamma) \cdot \mathbb{E}_{\varsigma_\pi}[(\Delta Q^{(l)})^2]$. In addition, from the fact that $\mathbb{E}_{\varsigma_\pi}[Q(\mathcal{P}^* - \mathcal{P})Q] = 0$ for all $Q \in L^2(\varsigma_\pi)(\mathcal{S} \times \mathcal{A})$, it holds that

$$
\begin{aligned}
&-\mathbb{E}_{\varsigma_\pi}[\Delta Q^{(l+1)}(\mathcal{P}^* - \mathcal{P})\Delta Q^{(l)}] \\
&= -\mathbb{E}_{\varsigma_\pi}[\Delta Q^{(l+1)}(\mathcal{P}^* - \mathcal{P})\Delta Q^{(l)}] + \mathbb{E}_{\varsigma_\pi}[\Delta Q^{(l+1)}(\mathcal{P}^* - \mathcal{P})\Delta Q^{(l+1)}] \\
&= \mathbb{E}_{\varsigma_\pi}[\Delta Q^{(l+1)}(\mathcal{P}^* - \mathcal{P})(Q^{(l+1)} - Q^{(l)})] \\
&\leq \frac{1}{2} \cdot \frac{2(1 - \gamma)}{\gamma} \mathbb{E}_{\varsigma_\pi}[(\Delta Q^{(l+1)})^2] + \frac{1}{2} \cdot \frac{\gamma}{2(1 - \gamma)} \|(\mathcal{P}^* - \mathcal{P})(Q^{(l+1)} - Q^{(l)})\|_{\varsigma_\pi,2}^2 \\
&\leq \frac{1 - \gamma}{\gamma} \mathbb{E}_{\varsigma_\pi}[(\Delta Q^{(l+1)})^2] + \frac{\gamma}{1 - \gamma} \|\mathcal{P}(Q^{(l+1)} - Q^{(l)})\|_{\varsigma_\pi,2}^2 \\
&\leq \frac{1 - \gamma}{\gamma} \mathbb{E}_{\varsigma_\pi}[(\Delta Q^{(l+1)})^2] + \frac{\gamma}{1 - \gamma} \mathbb{E}_{\varsigma_\pi}[(Q^{(l+1)} - Q^{(l)})^2],
\end{aligned}
\tag{17}
$$

where the first inequality follows from Young's inequality with an arbitrary constant $\frac{2(1-\gamma)}{\gamma} > 0$, and the last one follows from Lemma 7. Combining Eq. (15), (16), (17), and (14), we obtain that

$$
\begin{aligned}
&-\mathbb{E}_{\varsigma_\pi}[(Q^{(l)} - \mathcal{T}Q^{(l)}) \cdot (Q^{(l+1)} - Q_\pi)] \\
&= -\mathbb{E}_{\varsigma_\pi}[\Delta Q^{(l+1)}(I - \gamma \mathcal{P})\Delta Q^{(l)}] \\
&= -\frac{1}{2}\mathbb{E}_{\varsigma_\pi}[\Delta Q^{(l+1)}(I - \gamma \mathcal{P})\Delta Q^{(l+1)}] - \frac{1}{2}\mathbb{E}_{\varsigma_\pi}[\Delta Q^{(l)}(I - \gamma \mathcal{P})\Delta Q^{(l)}] \\
&\quad - \frac{\gamma}{2}\mathbb{E}_{\varsigma_\pi}[\Delta Q^{(l+1)}(\mathcal{P}^* - \mathcal{P})\Delta Q^{(l)}] \\
&\quad + \frac{1}{2}\mathbb{E}_{\varsigma_\pi}[(\Delta Q^{(l+1)} - \Delta Q^{(l)})(I - \gamma \mathcal{P})(\Delta Q^{(l+1)} - \Delta Q^{(l)})] \\
&\leq -\frac{1 - \gamma}{2}\mathbb{E}_{\varsigma_\pi}[(\Delta Q^{(l)})^2] + \frac{1}{2(1 - \gamma)}\mathbb{E}_{\varsigma_\pi}[(Q^{(l+1)} - Q^{(l)})^2].
\end{aligned}
\tag{18}
$$

Plugging Eq. (18) into Eq. (13), we obtain the following upper-bound of Eq. (13) as

$$
\begin{aligned}
\mathcal{L}_l[q_\pi] - \mathcal{L}_l[q^{(l+1)}] \leq &\frac{\gamma(2 - \gamma)}{2(1 - \gamma)}\mathbb{E}_{\varsigma_\pi}[(\Delta Q^{(l)})^2] \\
&+ \lambda_{\mathrm{TD}} \cdot D_{\mathrm{KL}}(q_\pi \| \nu) - \lambda_{\mathrm{TD}} \cdot D_{\mathrm{KL}}(q^{(l+1)} \| \nu),
\end{aligned}
\tag{19}
$$

In what follows, we give a lower bound on the difference for majorization objectives using the strong convexity of $L_l$. From the definition of $\mathcal{L}_l$ in Eq. (12), it holds that

$$\frac{\delta \mathcal{L}_l}{\delta q}[q](\omega) = \mathbb{E}_{\varsigma_\pi}[(Q^{(l)} - \mathcal{T}Q^{(l)}) \cdot h_\omega] + \frac{1}{1-\gamma}\mathbb{E}_{\varsigma_\pi}[(Q^{(l)} - Q_q) \cdot h_\omega] + \lambda_{\text{TD}} \cdot \ln \frac{q}{\nu}. \quad (20)$$

In what we follow, we control the error induced by the difference between the last iterate of inner-loop dynamics, $q^{(l+1)}$, and the optimal distribution of $\mathcal{L}^{(l)}$, $q_*^{(l+1)}$. It holds from Eq. (20) that

$$-\int \frac{\delta \mathcal{L}_l}{\delta q}[q^{(l+1)}](\mathrm{d}q_\pi - \mathrm{d}q^{(l+1)}) - \lambda_{\text{TD}} \cdot D_{\text{KL}}(q_\pi \| q^{(l+1)})$$

$$= \mathbb{E}_{\varsigma_\pi}[(Q^{(l)} - \mathcal{T}Q^{(l)}) \cdot (Q^{(l+1)} - Q_\pi)] + \frac{1}{1-\gamma}\mathbb{E}_{\varsigma_\pi}[(Q^{(l+1)} - Q^s) \cdot (Q^{(l+1)} - Q_\pi)]$$

$$- \lambda_{\text{TD}} \cdot \int \ln \frac{q^{(l+1)}}{\nu}(\mathrm{d}q_\pi - \mathrm{d}q^{(l+1)}) - \lambda_{\text{TD}} \cdot D_{\text{KL}}(q_\pi \| q^{(l+1)})$$

$$= \mathbb{E}_{\varsigma_\pi}[(Q^{(l)} - \mathcal{T}Q^{(l)}) \cdot (Q^{(l+1)} - Q_\pi)] + \frac{1}{1-\gamma}\mathbb{E}_{\varsigma_\pi}[(Q^{(l+1)} - Q^{(l)}) \cdot (Q^{(l+1)} - Q_\pi)] \quad (21)$$

$$- \lambda_{\text{TD}} \cdot D_{\text{KL}}(q_\pi \| \nu) + \lambda_{\text{TD}} \cdot D_{\text{KL}}(q^{(l+1)} \| \nu).$$

Plugging Eq. (21) into Eq. (13), we have

$$\mathcal{L}_l[q_\pi] - \mathcal{L}_l[q^{(l+1)}] = -\mathbb{E}_{\varsigma_\pi}[(Q^{(l)} - \mathcal{T}Q^{(l)}) \cdot (Q^{(l+1)} - Q_\pi)]$$

$$+ \frac{1}{2(1-\gamma)}\mathbb{E}_{\varsigma_\pi}[(Q^{(l)} - Q_\pi)^2] - \frac{1}{2(1-\gamma)}\mathbb{E}_{\varsigma_\pi}[(Q^{(l)} - Q^{(l+1)})^2]$$

$$+ \lambda_{\text{TD}} \cdot D_{\text{KL}}(q_\pi \| \nu) - \lambda_{\text{TD}} \cdot D_{\text{KL}}(q^{(l+1)} \| \nu)$$

$$= \frac{1}{2(1-\gamma)}\mathbb{E}_{\varsigma_\pi}[(Q^{(l)} - Q_\pi)^2] - \frac{1}{2(1-\gamma)}\mathbb{E}_{\varsigma_\pi}[(Q^{(l)} - Q^{(l+1)})^2]$$

$$+ \frac{1}{1-\gamma} \cdot \mathbb{E}_{\varsigma_\pi}[(Q^{(l+1)} - Q^{(l)}) \cdot (Q^{(l+1)} - Q_\pi)]$$

$$+ \int \frac{\delta \mathcal{L}_l}{\delta q}[q^{(l+1)}](\mathrm{d}q_\pi - \mathrm{d}q^{(l+1)}) - \lambda_{\text{TD}} \cdot D_{\text{KL}}(q_\pi \| q^{(l+1)})$$

$$= \frac{1}{2(1-\gamma)}\mathbb{E}_{\varsigma_\pi}[(Q^{(l+1)} - Q_\pi)^2] \quad (22)$$

$$+ \int \frac{\delta \mathcal{L}_l}{\delta q}[q^{(l+1)}](\mathrm{d}q_\pi - \mathrm{d}q^{(l+1)}) - \lambda_{\text{TD}} \cdot D_{\text{KL}}(q_\pi \| q^{(l+1)})$$

Note that if the output $q^{(l+1)}$ of the inner algorithm were the optimal solution $q_*^{(l)}$ of $L_l$, then in Eq. (22), $\int \frac{\delta \mathcal{L}_l}{\delta q}[q^{(l+1)}](\mathrm{d}q_\pi - \mathrm{d}q^{(l+1)}) - \lambda_{\text{TD}} \cdot D_{\text{KL}}(q_\pi \| q^{(l+1)})$ would be zero, and Eq. (22) would be the optimality condition of $q_*^{(l)}$ itself. In the sequel, we obtain the error bound by establishing Lemma 5.

**Lemma 5** (Inner-Loop Error Bound). *Under assumptions of Proposition 5, for any $l \in \mathbb{N}, s > 0$, we have*

$$-\int \frac{\delta \mathcal{L}_l}{\delta q}[q_s](\mathrm{d}q_\pi - \mathrm{d}q_s) - \lambda_{\text{TD}} \cdot D_{\text{KL}}(q_\pi \| q_s)$$

$$= \frac{2R}{1-\gamma}\left(\mathbb{E}_{\varsigma_\pi}[(Q_{q_s} - q_*^{(l+1)})^2]\right)^{\frac{1}{2}} + \lambda_{\text{TD}} \cdot D_{\text{KL}}(q_s \| q_*^{(l+1)}).$$

*Proof.* See Appendix C.3 for a detailed proof. □

Recall that the inner dynamics is stopped at the time $s = S > 0$ when we set the next outer iterate $q^{(l+1)}$ as that $q^{(l+1)} = q_S$. Combining Eq. (19), Eq. (22), and Eq. 5, we finish the proof of Lemma 1. □

## C.2 PROOF OF THEOREM 1

*Proof.* Before jumping to the proof of Theorem 1, we evaluate the mean-squared error between the Q-function $Q_{q_s}$ induced by $q_s$ and the global optimal Q-function $q_*^{(l+1)} = Q_{q^*}$ over the $L^2$-norm. In the sequel, we provide the following convergence lemma about the mean squared error of Q-functions.

**Lemma 6** (Linear Convergence of the Mean Squared Error of Q-functions). *Under the same assumption of Theorem 1, for $l \in \mathbb{N}$ we have*

$$\mathbb{E}_{\varsigma_\pi}[(Q^{(l+1)} - q_*^{(l+1)})^2] \leq \frac{4(3 - 2\gamma)R^4}{1 - \gamma} \cdot \exp(-2\alpha\lambda_{\text{TD}}S).$$

*where we denote $q_*^{(l+1)} = Q_{q_*^{(l)}}$ with the global optimal distribution $q_*^{(l+1)}$ of the inner objective $\mathcal{L}_l$ and the definition of each variable follows that of Proposition 5.*

*Proof.* For any parameter distributions $q, q' \in \mathcal{P}_2$, we first upper bound the Q-function difference with the Wasserstein distance.

$$
\begin{aligned}
(Q_q(x) - Q_{q'}(x))^2 &= \left( \int h_\omega(x)(\text{d}q(\omega) - \text{d}q'(\omega)) \right)^2 \\
&\leq R^2 \cdot \|q - q'\|_1^2 \\
&\leq 2R^2 \cdot D_{\text{KL}}(q\|q'),
\end{aligned}
\tag{23}
$$

where $R > 0$ is an absolute constant defined in Assumption 1 and the last inequality follows from Pinsker's inequality. Combining eq. (23) and Proposition 5, we obtain that

$$
\begin{aligned}
\mathbb{E}_{\varsigma_\pi}[(Q_{q_s} - q_*^{(l+1)})^2] &\leq 2R^2 \cdot D_{\text{KL}}(q_s\|q_*^{(l+1)}) \\
&= 2R^2 \exp(-2\alpha\lambda_{\text{TD}}s) \cdot \left( \mathcal{L}_l[q_0] - \mathcal{L}_l[q_*^{(l+1)}] \right).
\end{aligned}
$$

In order to control the right-hand side of the inequality above, given $\mathcal{L}_l > 0$ for any condition by the definition, we only take care of the boundness of $\mathcal{L}_l[q_0]$. It holds from Assumption 1, 2, that

$$
\begin{aligned}
\mathcal{L}_l[q_0] &= \mathbb{E}_{\varsigma_\pi}[(Q^{(l)} - \mathcal{T}Q^{(l)}) \cdot (Q_0 - Q_\pi)] \\
&\quad + \frac{1}{2(1 - \gamma)} \mathbb{E}_{\varsigma_\pi}[(Q^{(l)} - Q_0)^2] + \lambda_{\text{TD}} \cdot D_{\text{KL}}(q_0\|\nu) \\
&\leq 4R^2 + \frac{1}{2(1 - \gamma)} 4R^2 \\
&= \frac{2(3 - 2\gamma)R^2}{1 - \gamma},
\end{aligned}
$$

which concludes the proof. □

By Lemma 1, we have

$$
\begin{aligned}
\mathbb{E}_{\varsigma_\pi}[(Q^{(l+1)} - Q_\pi)^2] &\leq \frac{\gamma(2 - \gamma)}{(1 - \gamma)^2} \mathbb{E}_{\varsigma_\pi}\left[ (\Delta Q^{(l)})^2 - (\Delta Q^{(l+1)})^2 \right] \\
&\quad + \frac{4R}{(1 - \gamma)^2} (\mathbb{E}_{\varsigma_\pi}[(Q^{(l+1)} - q_*^{(l+1)})^2])^{\frac{1}{2}} \\
&\quad + \frac{2\lambda_{\text{TD}}}{1 - \gamma} \cdot D_{\text{KL}}(q^{(l+1)}\|q_*^{(l+1)}) + \frac{2\lambda_{\text{TD}}}{1 - \gamma} D_{\text{KL}}(q_\pi\|\nu).
\end{aligned}
$$

Combining Lemma 6 and Proposition 5, by the same argument of the proof of Lemma 6, it holds that

$$
\begin{aligned}
\mathbb{E}_{\varsigma_\pi}[(Q^{(l+1)} - Q_\pi)^2] &\leq \frac{\gamma(2 - \gamma)}{(1 - \gamma)^2} \mathbb{E}_{\varsigma_\pi}\left[ (\Delta Q^{(l)})^2 - (\Delta Q^{(l+1)})^2 \right] \\
&\quad + \frac{8(3 - 2\gamma)^{\frac{1}{2}}R^3}{(1 - \gamma)^{\frac{3}{2}}} \cdot \exp(-\alpha\lambda_{\text{TD}}S) \\
&\quad + \frac{8(3 - 2\gamma)\lambda_{\text{TD}}R^4}{(1 - \gamma)^2} \cdot \exp(-2\alpha\lambda_{\text{TD}}S) + \frac{2\lambda_{\text{TD}}M}{1 - \gamma}.
\end{aligned}
\tag{24}
$$

Telescoping (24) for $s = 0, \ldots, T_{\text{TD}} - 1$, we obtain

$$
\begin{aligned}
\frac{1}{T_{\text{TD}}} \sum_{s=1}^{T_{\text{TD}}} \mathbb{E}_{\varsigma_\pi}[(Q^{(l)} - Q_\pi)^2] \leq & \frac{\gamma(2 - \gamma)}{(1 - \gamma)^2 T_{\text{TD}}} \mathbb{E}_{\varsigma_\pi} \left[ (\Delta Q^{(0)})^2 - (\Delta Q^{(T_{\text{TD}})})^2 \right] \\
& + \frac{8(3 - 2\gamma)^{\frac{1}{2}} R^3}{(1 - \gamma)^{\frac{3}{2}}} \cdot \exp\left( -\alpha \lambda_{\text{TD}} S \right) \\
& + \frac{8(3 - 2\gamma) \lambda_{\text{TD}} R^4}{(1 - \gamma)^2} \cdot \exp\left( -2\alpha \lambda_{\text{TD}} S \right) + \frac{2\lambda_{\text{TD}} M}{1 - \gamma} \\
\leq & \frac{\gamma(2 - \gamma)}{(1 - \gamma)^2 T_{\text{TD}}} \mathbb{E}_{\varsigma_\pi} \left[ (\Delta Q^{(0)})^2 \right] \\
& + \frac{8(3 - 2\gamma)^{\frac{1}{2}} R^3}{(1 - \gamma)^{\frac{3}{2}}} \cdot \exp\left( -\alpha \lambda_{\text{TD}} S \right) \\
& + \frac{8(3 - 2\gamma) \lambda_{\text{TD}} R^4}{(1 - \gamma)^2} \cdot \exp\left( -2\alpha \lambda_{\text{TD}} S \right) + \frac{2\lambda_{\text{TD}} M}{1 - \gamma}.
\end{aligned}
$$

Recall that $R$ is the neural network radius satisfying $Q \leq R_r \leq R$ for any Q-function $Q$, we have

$$
\mathbb{E}_{\varsigma_\pi}[(\Delta Q^{(0)})^2] = \mathbb{E}_{\varsigma_\pi}[(Q^{(0)} - Q_\pi)^2] \leq 4R^2.
$$

Therefore, we have

$$
\begin{aligned}
\frac{1}{T_{\text{TD}}} \sum_{s=1}^{T_{\text{TD}}} \mathbb{E}_{\varsigma_\pi}[(Q^{(l)} - Q_\pi)^2] \leq & \frac{4\gamma(2 - \gamma) R^2}{(1 - \gamma)^2 T_{\text{TD}}} + \frac{8(3 - 2\gamma)^{\frac{1}{2}} R^3}{(1 - \gamma)^{\frac{3}{2}}} \cdot \exp\left( -\alpha \lambda_{\text{TD}} S \right) \\
& + \frac{8(3 - 2\gamma) \lambda_{\text{TD}} R^4}{(1 - \gamma)^2} \cdot \exp\left( -2\alpha \lambda_{\text{TD}} S \right) + \frac{2\lambda_{\text{TD}} M}{1 - \gamma}
\end{aligned}
$$

which concludes the proof of Theorem 1. $\qquad\square$

### C.3  PROOF OF LEMMA 5

*Proof.* We first present some lemmas on convergence properties. In specific, we prove the convergence of the parameter distribution $q_s$ to the global optimal distribution $q^*$ in the inner-loop MFLD and also Using the two convergence lemmas above, we evaluate the error derived from the inner-loop algorithm.

$$
\begin{aligned}
& -\int \frac{\delta \mathcal{L}_l}{\delta q}[q_s](\mathrm{d}q_\pi - \mathrm{d}q_s) - \lambda_{\text{TD}} \cdot D_{\text{KL}}(q_\pi \| q_s) \\
& = -\int \left( \frac{\delta \mathcal{L}_l}{\delta q}[q_s] - \frac{\delta \mathcal{L}_l}{\delta q}[q_*^{(l+1)}] \right) (\mathrm{d}q_\pi - \mathrm{d}q_s) - \lambda_{\text{TD}} \cdot D_{\text{KL}}(q_\pi \| q_s), \quad (25)
\end{aligned}
$$

where the last equality follows from the optimal condition with the stationary point $q_*^{(l+1)}$ as

$$
\frac{\delta \mathcal{L}_l}{\delta q}[q_*^{(l+1)}] = \text{const.}
$$

For the first term on the right-hand side of Eq. (25), the difference of the first-variations of $\mathcal{L}_l$ satisfies from the definition in Eq. (7) that

$$
\frac{\delta \mathcal{L}_l}{\delta q}[q_s] - \frac{\delta \mathcal{L}_l}{\delta q}[q_*^{(l+1)}] = \frac{1}{1 - \gamma} \mathbb{E}_{\varsigma_\pi} \left[ (q_*^{(l+1)} - Q_{q_s}) \cdot h_\omega \right] + \lambda_{\text{TD}} \cdot \ln \frac{q_s}{q_*^{(l+1)}}. \quad (26)
$$

Plugging Eq. (26) into Eq. (25), we have

$$
\begin{aligned}
-\int &\frac{\delta \mathcal{L}_l}{\delta q}[q_s](\mathrm{d}q_\pi - \mathrm{d}q_s) - \lambda_{\mathrm{TD}} \cdot D_{\mathrm{KL}}(q_\pi \| q_s) \\
&= \frac{1}{1-\gamma} \mathbb{E}_{\varsigma_\pi} \left[ (q_*^{(l+1)} - Q_{q_s}) \cdot (Q_{q_s} - Q_\pi) \right] \\
&\quad - \lambda_{\mathrm{TD}} \cdot \int \ln \frac{q_s}{q_*^{(l+1)}} (\mathrm{d}q_\pi - \mathrm{d}q_s) - \lambda_{\mathrm{TD}} \cdot D_{\mathrm{KL}}(q_\pi \| q_s) \\
&\leq \frac{1}{1-\gamma} \mathbb{E}_{\varsigma_\pi} \left[ (Q_{q_s} - q_*^{(l+1)})^2) \right]^{\frac{1}{2}} \cdot \| Q_{q_s} - Q_\pi \|_{\varsigma_\pi, 2} \\
&\quad + \lambda_{\mathrm{TD}} \cdot D_{\mathrm{KL}}(q_s \| q_*^{(l+1)}) - \lambda_{\mathrm{TD}} \cdot D_{\mathrm{KL}}(q_\pi \| q_*^{(l+1)}) \\
&\leq \frac{2R}{1-\gamma} \mathbb{E}_{\varsigma_\pi} \left[ (Q_{q_s} - q_*^{(l+1)})^2) \right]^{\frac{1}{2}} + \lambda_{\mathrm{TD}} \cdot D_{\mathrm{KL}}(q_s \| q_*^{(l+1)}) - \lambda_{\mathrm{TD}} \cdot D_{\mathrm{KL}}(q_\pi \| q_*^{(l+1)}) \\
&\leq \frac{2R}{1-\gamma} \mathbb{E}_{\varsigma_\pi} \left[ (Q_{q_s} - q_*^{(l+1)})^2) \right]^{\frac{1}{2}} + \lambda_{\mathrm{TD}} \cdot D_{\mathrm{KL}}(q_s \| q_*^{(l+1)}),
\end{aligned}
$$

where the second inequality follows from Assumption 1 and 2.

$\square$

# D  MEAN-FIELD LANGEVIN POLICY GRADIENT

## D.1  PROOF OF PROPOSITION 1

*Proof.* By Proposition 4, we have for all $\pi_\rho$ that

$$
\begin{aligned}
\mathrm{d}J[\rho] &= \mathbb{E}_{\nu_{\pi_\rho}} \left[ \int \mathrm{d}\pi_\rho(\mathrm{d}a) \cdot Q_{\pi_\rho}(a) \right] \\
&= \mathbb{E}_{\nu_{\pi_\rho}} \left[ \int \left( -\mathrm{d}f_\rho(a) + \int \pi_\rho(\mathrm{d}a') \mathrm{d}f_\rho(a') \right) \pi_\rho(\mathrm{d}a) \cdot Q_{\pi_\rho}(a) \right] \\
&= \mathbb{E}_{\nu_{\pi_\rho}} \left[ -\int \pi_\rho(\mathrm{d}a) \mathrm{d}f_\rho(a) \cdot Q_{\pi_\rho}(a) + \left( \int \pi_\rho(\mathrm{d}a') \mathrm{d}f_\rho(a') \right) \cdot \left( \int \pi_\rho(\mathrm{d}a) Q_{\pi_\rho}(a) \right) \right] \\
&= -\mathbb{E}_{\nu_{\pi_\rho}} \left[ \int \pi_\rho(\mathrm{d}a) \mathrm{d}f_\rho(a) \cdot \left( Q_{\pi_\rho}(a) - \int \pi_\rho(\mathrm{d}a') Q_{\pi_\rho}(a') \right) \right] \\
&= -\mathbb{E}_{\sigma_{\pi_\rho}} \left[ \mathrm{d}f_\rho \cdot A_{\pi_\rho} \right] \\
&= \int \mathrm{d}\rho(\mathrm{d}\theta) \mathbb{E}_{\sigma_{\pi_\rho}} \left[ -h_\theta \cdot A_{\pi_\rho} \right].
\end{aligned}
$$

From the first-variation of $J[\rho]$ is written from the Definition 1, it holds that

$$
\mathrm{d}J[\rho] = \int \mathrm{d}\rho(\mathrm{d}\theta) \frac{\delta J}{\delta \rho}[\rho].
$$

Comparing Equation D.1 and Equation D.1, we obtain Eq. (4). $\square$

## D.2  PROOF OF LEMMA 2

*Proof.* First of all, we define the PGD of $\mathcal{F}$ around $\rho_t$ by $\hat{\rho}_t \propto \exp \left( -\frac{1}{\lambda} \frac{\delta F}{\delta \rho}[\rho_t] \right)$. We calculate the time derivative of $\mathcal{F}$:

$$
\frac{\mathrm{d}}{\mathrm{d}t} \mathcal{F}[\rho_t] = \int \frac{\delta \mathcal{F}}{\delta \rho}[\rho_t] \partial_t \rho_t(\mathrm{d}\theta). \tag{27}
$$

Since we have $\mathcal{F}[\rho] = F[\rho] + \lambda \cdot \mathrm{Ent}[\rho]$, it holds that

$$
\begin{aligned}
\frac{\delta \mathcal{F}}{\delta \rho}[\rho] =& \frac{\delta F}{\delta \rho}[\rho] + \lambda \cdot \ln \rho \\
=& - \lambda \cdot \ln \exp \left( - \frac{1}{\lambda} \frac{\delta F}{\delta \rho}[\rho] \right) + \lambda \cdot \ln \rho \\
=& \lambda \cdot \ln \frac{\rho}{\hat{\rho}} - \lambda \ln \cdot Z[\hat{\rho}],
\end{aligned}
\tag{28}
$$

where we denote by $Z[\cdot]$ a normalization constant. In addition to that, from the definitions we have the following Fokker-Planck equation about the time evolution of $\rho_t$:

$$
\begin{aligned}
\partial_t \rho_t =& \lambda \cdot \Delta \rho_t + \nabla \cdot (\rho_t \cdot \nabla g_t[\rho_t]) \\
=& \lambda \cdot \nabla \left( \rho_t \cdot \nabla \ln \frac{\rho_t}{\tilde{\rho}_t} \right),
\end{aligned}
\tag{29}
$$

where $\tilde{\rho}_t$ is the approximation of $\hat{\rho}_t$. Plugging Eq. (28) and (29) into Eq. (27), it holds that

$$
\begin{aligned}
\frac{\mathrm{d}}{\mathrm{d}t} \mathcal{F}[\rho_t] =& \lambda^2 \int \ln \frac{\rho_t}{\hat{\rho}_t} \nabla \cdot \left( \rho_t \cdot \nabla \ln \frac{\rho_t}{\tilde{\rho}_t} \right) \mathrm{d}\theta \\
=& - \lambda^2 \int \rho_t(\mathrm{d}\theta) \left( \nabla \ln \frac{\rho_t}{\hat{\rho}_t} \right)^\top \left( \nabla \ln \frac{\rho_t}{\tilde{\rho}_t} \right) \\
\leq& - \frac{\lambda^2}{2} \int \rho_t(\mathrm{d}\theta) \left( \left\| \nabla \ln \frac{\rho_t}{\hat{\rho}_t} \right\|_2^2 - \left\| \nabla \ln \frac{\hat{\rho}_t}{\tilde{\rho}_t} \right\|_2^2 \right).
\end{aligned}
\tag{30}
$$

For the first term on the right-hand side of Eq. (30), it holds from LSI of $\hat{\rho}_t$ with the LSI constant $\alpha > 0$ which is given in Proposition 2 that

$$
\begin{aligned}
- \frac{\lambda^2}{2} \int \rho_t(\mathrm{d}\theta) \left\| \nabla \ln \frac{\rho_t}{\hat{\rho}_t} \right\|_2^2 =& - \frac{\lambda^2}{2} \mathrm{I}(\rho_t \| \hat{\rho}_t) \\
\leq& - \alpha \lambda^2 \cdot D_{\mathrm{KL}}(\rho_t \| \hat{\rho}_t).
\end{aligned}
$$

Note that $\alpha$ depends on $\lambda$ at the order $\mathcal{O}(\exp(-1/\lambda))$. See Proposition 2 for the detail of the construction of the LSI constant. Furthermore, we obtain, for all $\beta > 0$, that

$$
\begin{aligned}
- \lambda \cdot D_{\mathrm{KL}}(\rho_t \| \hat{\rho}_t) =& - \lambda \int \rho_t(\mathrm{d}\theta) \ln \frac{\rho_t}{\hat{\rho}_t} \\
=& \lambda \int \ln \frac{\rho_t}{\hat{\rho}_t} (\beta \cdot \hat{\rho}_t - \rho_t) \mathrm{d}\theta + \lambda \beta \cdot D_{\mathrm{KL}}(\hat{\rho}_t \| \rho_t).
\end{aligned}
$$

Considering the minimum of $\lambda \int \ln \frac{\rho_t}{\hat{\rho}_t} (\beta \cdot \tilde{\rho} - \rho_t) \mathrm{d}\theta + \lambda \beta \cdot D_{\mathrm{KL}}(\tilde{\rho} \| \rho_t) = \lambda \beta \cdot D_{\mathrm{KL}}(\tilde{\rho} \| \hat{\rho}_t) - \lambda \cdot D_{\mathrm{KL}}(\rho_t \| \hat{\rho}_t)$ for all $\tilde{\rho} \in \mathcal{P}_2$, we obtain that

$$
\begin{aligned}
- \lambda \cdot D_{\mathrm{KL}}(\rho_t \| \hat{\rho}_t) =& \lambda \int \ln \frac{\rho_t}{\hat{\rho}_t} (\beta \cdot \hat{\rho}_t - \rho_t) \mathrm{d}\theta + \lambda \beta \cdot D_{\mathrm{KL}}(\hat{\rho}_t \| \rho_t) \\
=& \min_{\tilde{\rho} \in \mathcal{P}_2} \left\{ \lambda \int \ln \frac{\rho_t}{\hat{\rho}_t} (\beta \cdot \tilde{\rho} - \rho_t) \mathrm{d}\theta + \lambda \beta \cdot D_{\mathrm{KL}}(\tilde{\rho} \| \rho_t) \right\} \\
\leq& \lambda \int \ln \frac{\rho_t}{\hat{\rho}_t} (\beta \cdot \tilde{\rho} - \rho_t) \mathrm{d}\theta + \lambda \beta \cdot D_{\mathrm{KL}}(\tilde{\rho} \| \rho_t) \\
=& \int \mathbb{E}_{\sigma_t} \left[ A_{\pi_t} \cdot h_\theta \right] (\beta \cdot \tilde{\rho} - \rho_t) \mathrm{d}\theta \\
& + \lambda \int \ln \frac{\rho_t}{\nu} (\beta \cdot \tilde{\rho} - \rho_t) \mathrm{d}\theta + \lambda \beta \cdot D_{\mathrm{KL}}(\tilde{\rho} \| \rho_t),
\end{aligned}
\tag{31}
$$

where $\tilde{\rho} \in \mathcal{P}_2$ and $\beta > 0$ are arbitrary. On the right-hand side of Eq. (31), The first term holds that

$$
\begin{aligned}
\int \mathbb{E}_{\sigma_t} \left[ A_{\pi_t} h_\theta \right] (\beta \cdot \tilde{\rho} - \rho_t) \mathrm{d}\theta =& \mathbb{E}_{\sigma_t} \left[ A_{\pi_t} \left( \int \beta \cdot h_\theta \tilde{\rho}(\mathrm{d}\theta) - \int h_\theta \rho_t(\mathrm{d}\theta) \right) \right] \\
=& \mathbb{E}_{\sigma_t} \left[ A_{\pi_t} \left( \beta \cdot \tilde{f} - f_t \right) \right],
\end{aligned}
\tag{32}
$$

where $\tilde{f} = \int h_\theta \tilde{\rho}(\mathrm{d}\theta)$. Eq. (32) is expected to be upper bounded by the difference of the expected total rewards. By contrast, as for the KL part on the right-hand side of Eq. (31), we have

$$
\begin{aligned}
\int \ln \frac{\rho_t}{\nu} \left(\beta \cdot \tilde{\rho} - \rho_t\right) \mathrm{d}\theta + \beta \cdot D_{\mathrm{KL}}\left(\tilde{\rho}\|\rho_t\right) &= \int \ln \frac{\rho_t}{\nu} \left(\beta \cdot \tilde{\rho} - \rho_t\right)(\mathrm{d}\theta) + \beta \int \tilde{\rho}(\mathrm{d}\theta) \ln \frac{\tilde{\rho}}{\rho_t} \\
&= \beta \int \mathrm{d}\tilde{\rho} \ln \frac{\rho_t}{\nu} - \int \mathrm{d}\rho_t \ln \frac{\rho_t}{\nu} + \beta \int \mathrm{d}\tilde{\rho} \ln \frac{\tilde{\rho}}{\rho_t} \\
&= -\int \mathrm{d}\rho_t \ln \frac{\rho_t}{\nu} + \beta \int \mathrm{d}\tilde{\rho} \ln \frac{\tilde{\rho}}{\nu} \\
&= -D_{\mathrm{KL}}(\rho_t\|\nu) + \beta \cdot D_{\mathrm{KL}}(\tilde{\rho}\|\nu).
\end{aligned} \tag{33}
$$

Plugging Eq. (32) and Eq. (33) into Eq. (31), we have

$$
-\lambda \cdot D_{\mathrm{KL}}(\rho_t\|\hat{\rho}_t) \le \mathbb{E}_{\sigma_t}\left[A_{\pi_t} \cdot \left(\beta \cdot \tilde{f} - f_t\right)\right] - D_{\mathrm{KL}}(\rho_t\|\nu) + \beta \cdot D_{\mathrm{KL}}(\tilde{\rho}\|\nu).
$$

In the sequel, in order to upper bound the third term on the right-hand side of Eq. (30), we obtain the difference of advantage function as

$$
\begin{aligned}
\nabla \ln \frac{\hat{\rho}_t}{\tilde{\rho}_t} &= \left(-\frac{1}{\lambda}\nabla \frac{\delta \mathcal{F}}{\delta \rho}[\rho_t]\right) - \left(-\frac{1}{\lambda}\nabla g[\rho_t]\right) \\
&= \frac{1}{\lambda}\mathbb{E}_{\sigma_t}[\nabla h_\theta \cdot (A_t - A_{\pi_t})],
\end{aligned}
$$

where $A_t$ is the advantage function estimator given by the critic at time $t$. This term defines the policy evaluation error as

$$
\begin{aligned}
&\lambda^2 \int \rho_t(\mathrm{d}\theta) \left\|\nabla \ln \frac{\hat{\rho}_t}{\tilde{\rho}_t}\right\|_2^2 \\
&= \int \rho_t(\mathrm{d}\theta) \left\|\mathbb{E}_{\sigma_t}\left[\nabla h_\theta \cdot (A_{\pi_t} - A_t)\right]\right\|_2^2 \\
&= \int \rho_t(\mathrm{d}\theta) \left\|\mathbb{E}_{\sigma_t}\left[\nabla h_\theta \cdot (Q_t - Q_{\pi_t})\right] - \mathbb{E}_{\sigma_t}\left[\nabla h_\theta \cdot (V_t - V_{\pi_t})\right]\right\|_2^2 \\
&= \int \rho_t(\mathrm{d}\theta) \left\|\mathbb{E}_{\sigma_t}\left[\nabla h_\theta \cdot (Q_t - Q_{\pi_t})\right] - \mathbb{E}_{\nu_\pi}\left[\int \pi(\mathrm{d}a)\nabla h_\theta \cdot \left(\int \pi(\mathrm{d}a')(Q_t - Q_{\pi_t})\right)\right]\right\|_2^2 \\
&= \int \rho_t(\mathrm{d}\theta) \left\|\mathbb{E}_{\sigma_t}\left[\nabla h_\theta \cdot (Q_t - Q_{\pi_t})\right] - \mathbb{E}_{\nu_\pi}\left[\int \pi(\mathrm{d}a)(Q_t - Q_{\pi_t}) \cdot \left(\int \pi(\mathrm{d}a')\nabla h_\theta\right)\right]\right\|_2^2 \\
&= \int \rho_t(\mathrm{d}\theta) \left\|\mathbb{E}_{\sigma_t}\left[(Q_t - Q_\pi) \cdot \left(\nabla h_\theta - \int \pi(\mathrm{d}a')\nabla h_\theta\right)\right]\right\|_2^2 \\
&= \int \rho_t(\mathrm{d}\theta) \left\|\mathbb{E}_{\varsigma_t}\left[\frac{\mathrm{d}\sigma_t}{\mathrm{d}\varsigma_t} \cdot (Q_t - Q_{\pi_t}) \cdot \left(\nabla h_\theta - \int \pi(\mathrm{d}a')\nabla h_\theta\right)\right]\right\|_2^2,
\end{aligned}
$$

where we exchange $a$ with $a'$ at the forth equality and $\frac{\mathrm{d}\sigma_t}{\mathrm{d}\varsigma_t}$ is the Radon-Nikodim derivative between the state-action visitation measure $\sigma_t$ and the stationary state-action distribution $\varsigma_t$ corresponding to the same policy $\pi_t$. Since we assume that the neural network $h_\theta$ is $L_1$-Lipschitz continous in Assumption 1, we have $\|\nabla h_\theta\|_\infty \le L_1$. It holds that

$$
\begin{aligned}
&\int \rho_t(\mathrm{d}\theta) \left\|\mathbb{E}_{\varsigma_t}\left[\frac{\mathrm{d}\sigma_t}{\mathrm{d}\varsigma_t} \cdot (Q_t - Q_{\pi_t}) \cdot \left(\nabla h_\theta - \int \pi(\mathrm{d}a')\nabla h_\theta\right)\right]\right\|_2^2 \\
&\le \mathbb{E}_{\varsigma_t}\left[(Q_t - Q_{\pi_t})^2\right] \cdot \mathbb{E}_{\varsigma_t}\left[\left(\frac{\mathrm{d}\sigma_t}{\mathrm{d}\varsigma_t}\right)^2 \cdot \int \rho_t(\mathrm{d}\theta) \left\|\nabla h_\theta - \int \pi(\mathrm{d}a')\nabla h_\theta\right\|_2^2\right] \\
&\le 4L_2^2 \cdot \mathbb{E}_{\varsigma_t}\left[(Q_t - Q_{\pi_t})^2\right] \cdot \left\|\frac{\mathrm{d}\sigma_t}{\mathrm{d}\varsigma_t}\right\|_{\varsigma_t,2}^2,
\end{aligned} \tag{34}
$$

where the first inequality follows from Jansen's inequality. By plugging Eq. (32) and Eq. (33) into Eq. (31) and then plugging Eq. (31) and Eq. (46) into Eq. (30), we conclude that

$$\frac{\mathrm{d}}{\mathrm{d}t}\mathcal{F}[\rho_t] \leq \alpha\lambda \cdot \mathbb{E}_{\sigma_t}\left[A_{\pi_t} \cdot \left(\beta \cdot \tilde{f} - f_t\right)\right]$$
$$+ \alpha\lambda^2 \cdot \left(D_{\mathrm{KL}}\left(\tilde{\rho}\|\nu\right) - D_{\mathrm{KL}}\left(\rho_t\|\nu\right)\right) + 2L_1^2 \cdot \left\|\frac{\mathrm{d}\sigma_t}{\mathrm{d}\varsigma_t}\right\|_{\varsigma_t,2}^2 \cdot \mathbb{E}_{\varsigma_t}\left[\left(Q_t - Q_{\pi_t}\right)^2\right],$$

which concludes the proof of Lemma 2. $\qquad\square$

### D.3 PROOF OF THEOREM 2

*Proof.* It is well known that the expected total reward function $J[\pi]$ has non-convexity, which makes the optimization of the expected total rewards much more difficult. We first make use of a lemma to prove the one-point convexity of $J[\pi]$ at the global optimum $\pi^*$. This lemma is adapted from Kakade & Langford (2002).

**Proposition 3** (Expected Total Rewards Difference (Kakade & Langford, 2002)). *For all $\pi, \pi'$, it holds that*

$$(1-\gamma) \cdot (J[\pi'] - J[\pi]) = \mathbb{E}_{\sigma_{\pi'}}\left[A_\pi\right]$$

*where $\sigma_{\pi'}$ and $\nu_{\pi'}$ are the state-action visitation measure and the state visitation measure induced by policy $\pi'$, respectively.*

In our analysis, we utilize Proposition 3 as a one-point convexity of the expected total rewards to prove the global optimality of the stationary point of the MFLPG, which is provided by Lemma 2. In specific, we first evaluate the first term on the right-hand side of Eq. (10) with the performance difference. Let $\pi^*$ be the globally optimal policy of the expected total reward function $J$ and further define the globally optimal expected total reward by $J^* = J[\pi_*]$. By Proposition 3, it holds for all $t \in [0, T]$ that

$$J^* - J[\rho_t] = (1-\gamma)^{-1}\mathbb{E}_{\sigma_*}\left[A_{\pi_t}\right] = (1-\gamma)^{-1}\mathbb{E}_{\sigma_t}\left[\frac{\mathrm{d}\sigma_*}{\mathrm{d}\sigma_t} \cdot A_{\pi_t}\right], \tag{35}$$

where $\mathbb{E}_{\sigma_*}[\cdot] = \mathbb{E}_{\sigma_{\pi^*}}[\cdot]$, hereafter. As for the first term on the right-hand side of Eq. (10), combining Lemma 2 and Eq. (35), for all $\tilde{\rho} \in \mathcal{P}_2, \beta > 0$ we have

$$\mathbb{E}_{\sigma_t}\left[A_{\pi_t} \cdot \left(\beta \cdot \tilde{f} - f_t\right)\right] = -\left(J^* - J[\rho_t]\right)$$
$$+ \mathbb{E}_{\sigma_t}\left[A_{\pi_t} \cdot \left(\beta \cdot \tilde{f} - f_t\right)\right] + (1-\gamma)^{-1}\mathbb{E}_{\sigma_t}\left[\frac{\mathrm{d}\sigma_*}{\mathrm{d}\sigma_t} \cdot A_{\pi_t}\right]$$
$$= -\left(J^* - J[\rho_t]\right)$$
$$+ \mathbb{E}_{\sigma_t}\left[A_{\pi_t} \cdot \left(\beta \cdot \tilde{f} - f_t + (1-\gamma)^{-1}\frac{\mathrm{d}\sigma_*}{\mathrm{d}\sigma_t}\right)\right]. \tag{36}$$

As for the second term on the right-hand side in Eq. (36), we have

$$\mathbb{E}_{\sigma_t}\left[A_{\pi_t} \cdot \left(\beta \cdot \tilde{f} - f_t + (1-\gamma)^{-1}\frac{\mathrm{d}\sigma_*}{\mathrm{d}\sigma_t}\right)\right]$$
$$= \beta \cdot \langle A_{\pi_t}, \tilde{f}\rangle_{\sigma_t} - \langle A_{\pi_t}, f_t\rangle_{\sigma_t} + (1-\gamma)^{-1}\left\langle A_{\pi_t}, \frac{\mathrm{d}\sigma_*}{\mathrm{d}\sigma_t}\right\rangle_{\sigma_t}, \tag{37}$$

where $\langle \cdot, \cdot\rangle_{\sigma_t}$ denotes inner product which introduces $L_{\sigma_t,2}$ norm. Now we upper bound the second and third term on the right-hand side in Eq. (37). By Assumption 1, we obtain, from the fact that the neural network $h_{\rho_t}$ is bounded by $R > 0$, that

$$-\langle A_{\pi_t}, f_t\rangle_{\sigma_t} \leq |\langle A_{\pi_t}, f_t\rangle_{\sigma_t}|$$
$$\leq \|A_{\pi_t}\|_{\sigma_t,2} \cdot \|f_t\|_{\sigma_t,2}$$
$$\leq R \cdot \|A_{\pi_t}\|_{\sigma_t,2}, \tag{38}$$

where $R > 0$ is an absolute constant defined in Assumption 1. Meanwhile, we upper bound the third term on the right-hand side in Eq. (37) as

$$\left\langle A_{\pi_t}, \frac{\mathrm{d}\sigma_*}{\mathrm{d}\sigma_t} \right\rangle_{\sigma_t} \leq \|A_{\pi_t}\|_{\sigma_t,2} \cdot \left\| \frac{\mathrm{d}\sigma_*}{\mathrm{d}\sigma_t} \right\|_{\sigma_t,2}$$
$$\leq \kappa \cdot \|A_{\pi_t}\|_{\sigma_t,2}, \tag{39}$$

where the first inequality follows from Jansen's inequality, the second inequality follows from Assumption 4, and $\kappa > 0$ is an absolute constant defined in Assumption 4. By plugging Eq. (38) and (39) into Eq. (37), we have

$$\mathbb{E}_{\sigma_t}\left[ A_{\pi_t} \cdot \left( \beta \cdot \tilde{f} - f_t + (1-\gamma)^{-1}\frac{\mathrm{d}\sigma_*}{\mathrm{d}\sigma_t} \right) \right] \leq \beta \cdot \langle A_{\pi_t}, \tilde{f} \rangle_{\sigma_t} + (R+\kappa) \cdot \|A_{\pi_t}\|_{\sigma_t,2}$$
$$= \beta \cdot \left( \langle A_{\pi_t}, \tilde{f} \rangle_{\sigma_t} + \frac{R+\kappa}{\beta} \cdot \|A_{\pi_t}\|_{\sigma_t,2} \right). \tag{40}$$

From Assumption 5, there exists a function $f \in \mathcal{F}_{R,M}$ satisfying the following equation, where $M, B > 0$ are finite constants:

$$|\langle A_{\pi_t}, f \rangle_{\sigma_t}| \geq \frac{1}{B}\|A_{\pi_t}\|_{\sigma_t,2}.$$

If $\langle A_{\pi_t}, f \rangle_{\sigma_t} \leq 0$, then it holds that

$$\langle A_{\pi_t}, f \rangle_{\sigma_t} + \frac{1}{B}\|A_{\pi_t}\|_{\sigma_t,2} \leq 0.$$

By contrast, if $\langle A_{\pi_t}, f \rangle_{\sigma_t} > 0$, then we consider the following function $f'$. Recall that the second weight function $\beta(\cdot)$ is an odd function given by Assumption 1, from the definition of the neural networks we can easily create such a distribution $\rho'$ and $f' = \int h_\theta \mathrm{d}\rho'$ satisfying both $\langle A_{\pi_t}, f' \rangle_{\sigma_t} = -\langle A_{\pi_t}, f \rangle_{\sigma_t}$ and $D_{\mathrm{KL}}(\rho'\|\nu) = D_{\mathrm{KL}}(\rho\|\nu)$. That is, $f'$ satisfies $f' \in \mathcal{F}_{R,M}$ and also it holds that

$$\langle A_{\pi_t}, f' \rangle_{\sigma_t} + \frac{1}{B}\|A_{\pi_t}\|_{\sigma_t,2} \leq 0.$$

Therefore, from the above we have a certain function $\tilde{f} \in \mathcal{F}_{R,M}$ such that

$$\langle A_{\pi_t}, \tilde{f} \rangle_{\sigma_t} + \frac{1}{B}\|A_{\pi_t}\|_{\sigma_t,2} \leq 0. \tag{41}$$

Thus, we here set $\beta = B(R+\kappa) > 0$ and it holds by plugging Eq. (41) into Eq. (40) that

$$\mathbb{E}_{\sigma_t}\left[ A_{\pi_t} \cdot \left( \beta \cdot \tilde{f} - f_t + (1-\gamma)^{-1}\frac{\mathrm{d}\sigma_*}{\mathrm{d}\sigma_t} \right) \right] \leq 0. \tag{42}$$

As a result, Plugging Eq. (42) into Eq. (36), we have

$$\mathbb{E}_{\sigma_t}\left[ A_{\pi_t} \cdot \left( \beta \cdot \tilde{f} - f_t \right) \right] \leq -(J[\rho_*] - J[\rho_t]). \tag{43}$$

Furthermore, recall that the function $\tilde{f} \in \mathcal{F}_{R,M}$ satisfies from the definition in Eq. (8) that
$$D_{\mathrm{KL}}(\tilde{\rho}\|\nu) \leq M, \tag{44}$$

where $\tilde{f} = \int h_\theta \mathrm{d}\tilde{\rho}$. By plugging Eq. (44), (43), and $\beta = B(1+\kappa)$ into Eq. (10), we obtain that

$$\frac{\mathrm{d}}{\mathrm{d}t}\mathcal{F}[\rho_t] \leq -\alpha\lambda \cdot (J^* - J[\rho_t]) + \alpha\lambda^2 \cdot (BM(R+\kappa) - D_{\mathrm{KL}}(\rho_t\|\nu)) + 2L_2^2 \cdot \Delta_t$$
$$\leq -\alpha\lambda \cdot \left( \mathcal{F}[\rho_t] + J^* - \lambda BM(R+\kappa) - \frac{2L_2^2}{\alpha\lambda}\Delta_t \right). \tag{45}$$

As for the forth term on the right-hand side of Eq. (45), we upper bound the inner-loop error $\Delta_t$ from Assumption 4 and Theorem 1 with $T_{\mathrm{TD}} = \mathrm{O}(1/\lambda_{\mathrm{TD}})$ as

$$\Delta_t = \left\| \frac{\mathrm{d}\sigma_t}{\mathrm{d}\varsigma_t} \right\|_{\varsigma_t,2}^2 \cdot \mathbb{E}_{\varsigma_t}[(Q_t - Q_{\pi_t})^2]$$
$$\leq \iota \cdot \mathbb{E}_{\varsigma_t}[(Q_t - Q_{\pi_t})^2]$$
$$\leq \iota \cdot \mathcal{O}(\lambda_{\mathrm{TD}}). \tag{46}$$

In addition to that, set $\lambda_{\mathrm{TD}} = \mathcal{O}(\alpha\lambda^2)$ and define $\varepsilon(\lambda)$ as $\varepsilon(\lambda) = \lambda BM(R+\kappa) + \frac{2L_2^2}{\alpha\lambda}\Delta_t \geq 0$ for simplicity, then we have

$$\varepsilon(\lambda) = \mathcal{O}(\lambda).$$

Plugging Eq. (46) into Eq. (45), it holds that

$$\frac{\mathrm{d}}{\mathrm{d}t}\left(\mathcal{F}[\rho_t] + J^* - \varepsilon(\lambda)\right) \leq -\alpha\lambda \cdot \left(\mathcal{F}[\rho_t] + J^* - \varepsilon(\lambda)\right). \tag{47}$$

We obtain from a straightforward application of the Grönwall's inequality to Eq. (47) that

$$
\begin{aligned}
-J[\rho_t] + \lambda \cdot D_{\mathrm{KL}}(\rho_t\|\nu) + J^* - \varepsilon(\lambda) &\leq \exp(-2\alpha\lambda t)\cdot\left(\mathcal{F}[\rho_0] + J^* - \varepsilon(\lambda)\right)\\
&= \exp(-2\alpha\lambda t)\cdot\left(J^* - J[\rho_0] + \lambda\cdot D_{\mathrm{KL}}(\rho_0\|\nu) - \varepsilon(\lambda)\right)\\
&= \exp(-2\alpha\lambda t)\cdot\left(J^* - J[\rho_0] - \varepsilon(\lambda)\right)\\
&\leq \exp(-2\alpha\lambda t)\cdot\left(J^* - J[\rho_0]\right),
\end{aligned}
$$

where the second equality follows from the fact that $\rho_0 = \nu \sim \mathcal{N}(0, I_d)$. Therefore, we conclude that

$$J^* - J[\rho_T] \leq \exp(-2\alpha\lambda T)\cdot\left(J^* - J[\rho_0]\right) + \varepsilon(\lambda),$$

which concludes the proof of Theorem 2. $\qquad\square$

## E  AUXILIARY LEMMAS

This section is devoted to presenting the related propositions and lemmas used in the proof.

First of all, we present the policy gradient theorem presented by Sutton et al. (1999) as Proposition 4, which provides the gradient of the expected total reward function $J$ with a policy $\pi_\Theta$ parameterized by $\Theta$. Refer to the original paper for the proof.

**Proposition 4** (Policy Gradient Theorem (Sutton et al., 1999)). *For any MDP, it holds that*

$$\nabla J[\pi_\Theta] = \mathbb{E}_{\sigma_{\pi_\Theta}}\left[\int \nabla\pi_\Theta(\mathrm{d}a|s)\cdot Q_{\pi_\Theta}(s,a)\right],$$

*where $\sigma_{\pi_\theta}$ is the state visitation measure.*

Second, we provide the basic proposition to access the inner-loop convergence of MFLTD. For $l$-th outer step, we define the global optimal distribution of an inner-loop MFLD by $q_*^{(l+1)}$. It holds that

**Proposition 5** (Linear Convergence of Inner-Loop MFLD). *Under Assumption 1, 2, and 3, if we run the noisy gradient descent which is the inner-loop algorithm in Algorithm 2 for all $l$-step, we obtain for all $s \geq 0, l \in [0, T_{\mathrm{TD}} - 1]$ and $\lambda_{\mathrm{TD}} > 0$ that*

$$D_{\mathrm{KL}}(q_s\|q_*^{(l+1)}) \leq \mathcal{L}_l[q_s] - \mathcal{L}_l[q_*^{(l+1)}] \leq \exp\left(-2\alpha\lambda_{\mathrm{TD}}s\right)\cdot\left(\mathcal{L}_l[q_0] - \mathcal{L}_l[q_*^{(l+1)}]\right),$$

*where $\alpha$ is the LSI constant induced by $\lambda_{\mathrm{TD}}$.*

*Proof.* For the proof of Proposition 5, we apply Nitanda et al. (2022)'s convex optimization analysis. First of all, we prove the convexity of $L_l$ over the parameter distribution. From the definition of $L_l$, we can reformulate $L_l$ as

$$L_l[q] = \int U\mathrm{d}q + \left(\int V\mathrm{d}q\right)^2, \tag{48}$$

where $U, V$ do not depend on $q$ but only $\theta$. Eq. (48) results that the objective function $L_l$ is convex in terms of a functional with a probability distribution $q$ as a variable, where we define the convexity condition for functional $F$ and $q, q' \in \mathcal{P}_2$ as

$$F[q'] \geq F[q] + \int \frac{\delta F}{\delta q}[q](\theta)(\mathrm{d}q' - \mathrm{d}q).$$

Therefore, Noting that the MFLD used in the inner algorithm of the MFLTD is a simple Wasserstein gradient flow with the convex objective functional $L_l$, it holds from Theorem 1 in (Nitanda et al., 2022) that

$$\mathcal{L}_l[q_s] - \mathcal{L}_l[q_*^{(l+1)}] \leq \exp\left(-2\alpha\lambda_{\mathrm{TD}}s\right)\cdot\left(\mathcal{L}_l[q_0] - \mathcal{L}_l[q_*^{(l+1)}]\right),$$

where $\alpha$ is the LSI constant induced by $\lambda_{\mathrm{TD}}$, which is given in Proposition 2. In addition, the proof of the first inequality of the statement Proposition 5 follows from Proposition 1. in (Nitanda et al., 2022), from which we have

$$D_{\mathrm{KL}}(q_s \| q_*^{(l+1)}) \leq \mathcal{L}_l[q_s] - \mathcal{L}_l[q_*^{(l+1)}].$$

We finish the proof of Proposition 5. □

In the sequel, we introduce the following basic lemma about the norm of the transition operator, which is useful for a geometric property of the semi-gradient of the Bellman error.

**Lemma 7** (Transition Operator Norm). *Let the linear operator $\mathcal{P} : L^2(\varsigma_\pi)(\mathcal{S} \times \mathcal{A}) \to L^2(\varsigma_\pi)(\mathcal{S} \times \mathcal{A})$ be the transition operator satisfying $\mathcal{P}Q(s, a) = \int \mathrm{d}s' P(s'|s, a) \int \mathrm{d}a' \pi(a'|s')Q(s', a')$, $Q \in L^2(\varsigma_\pi)(\mathcal{S} \times \mathcal{A})$. Then the operator norm of $\mathcal{P}$ is no more than $1$.*

*Proof.* For all $Q(x) \in L^2(\varsigma_\pi)(\mathcal{S} \times \mathcal{A}), x = (s, a) \in \mathcal{S} \times \mathcal{A}$, we have that

$$
\begin{aligned}
\|\mathcal{P}Q(x)\|_{\varsigma_\pi, 2} =& \mathbb{E}_{x \sim \varsigma_\pi}\left[\left(\int \mathrm{d}(\pi \otimes P)(x'|x)Q(x')\right)^2\right] \\
\leq& \mathbb{E}_{x \sim \varsigma_\pi}\left[\int \mathrm{d}(\pi \otimes P)(x'|x)Q(x')^2\right] \\
=& \mathbb{E}_{x \sim \varsigma_\pi}\left[Q(x)^2\right] \\
=& \|Q(x)\|_{\varsigma_\pi, 2},
\end{aligned}
$$

where the inequality follows Jensen's inequality and the second equality follows the fact that $\varsigma_\pi$ is the stationary distribution under the transition probability. □

