# OpenReview forum: "Mean Field Langevin Actor-Critic: Faster Convergence and Global Optimality beyond Lazy Learning"
_ICLR.cc/2024/Conference — Submitted to ICLR 2024_

### Official Review · Reviewer_1VQp · 2023-10-16

**Soundness:** 2 fair
**Presentation:** 3 good
**Contribution:** 2 fair
**Rating:** 5
**Confidence:** 2

**Summary:**

To find the global optimal policy in reinforcement learning, this paper proposes the mean-field Langevin TD learning method (MFLTD) and mean-field Langevin policy gradient (MFLPG).
MFLTD converges to the true value function at a sublinear rate. MFLPG converges to the globally optimal policy of expected total reward at a linear convergence rate under KL-divergence regularization. This paper also provides the linear convergence guarantee for neural actor-critic algorithms with global optimality and feature learning.

**Strengths:**

1.	This paper proposes a new actor-critic approach.
2.	The theoretical analysis in this paper is sufficient.

**Weaknesses:**

1.	The reference part is missing in the main file.
2.	The global optimality should be analyzed in finite MDPs.
3.	The experiment is only on CartPole.
3.	There should be the experiment on the over-parameterized cases.

**Questions:**

1.	Why the networks are over-parameterized? In fact, more parameters can improve the performance of deep reinforcement learning. See OFE ([Ota et al., 2020] and [Ota et al., 2021]).
2.	Is the global optimality guaranteed in  finite MDPs? If some states are not reached, how can we guarantee global optimality?
3.	I would like the see the details of the feature learning in this work. Is there any encoded representation, like OFE and DREAMER?

---

> ### Author Response · Authors · 2023-11-22
> **Rebuttal by Authors**
>
> Thank you for your helpful comments. We address the technical comments below.
>
> **Regarding Finite MDPs**
>
> > The global optimality should be analyzed in finite MDPs. Is the global optimality guaranteed in finite MDPs?
>
> Finite MDPs represent a simpler subset within our MDP framework and are encompassed within our settings. Notably, our analysis stands out in its capacity to handle continuous state and action spaces. Additionally, in cases with a finite number of action spaces, the policies in MFLPG can be characterized by following a softmax function to probabilistically select actions. Research in reinforcement learning on MDPs, particularly policy gradient methods, has been actively pursued recently, and analyzing global optimality within such settings remains a significant topic ([Leathy et al. (2022)](https://proceedings.mlr.press/v162/leahy22a/leahy22a.pdf), Zhang et al. 2021, 2022).
>
> > If some states are not reached, how can we guarantee global optimality?
>
> In this study, we consistently regard the expected value on the visit distribution as what to get easily. Specifically, we derive results based solely on population updates, not relying on stochastic gradient descent, which aligns with recent studies ([Leathy et al. (2022)](https://proceedings.mlr.press/v162/leahy22a/leahy22a.pdf), Zhang et al. 2021, 2022). Moreover, the completeness of sampling subsets of states crucial for estimating the optimal policy is aggregated within the moment regularity condition, outlined in Assumption 4, related to the visitation distribution based on the optimal policy and the distribution at each time step.
>
> **Regarding the Motivation and Superiority of Experimentation in Section 5**
>
> > The experiment is only on CartPole.
>
> The computational experiments presented in Section 5 serve the explicit purpose of addressing two key objectives: (1) To experimentally demonstrate the necessity of analyzing network representation learning in the mean-field regime surpassing existing lazy-scaling theories like NTK concerning optimization and generalization capabilities, and (2) To showcase the practical utility of our newly proposed method, the Double-Loop MFLTD, by comparing it practically with the widely used, more basic TD(1). It's essential to note that our study's focal point lies in conducting convergence analyses of TD learning and PG methods in the mean-field regime, capturing feature learning without relying on lazy-scaling, and deriving rapid convergence rates under mild assumptions.
> In general, analytical studies of deep learning theory, especially within reinforcement learning, lack computational experiments or are confined to simplified reward structures, table transition settings, or classic toy models like Cart Pole. While showcasing the effectiveness of our algorithms in larger models or complex environments with vast state spaces would offer significant insights, considering the current stage with theoretical convergence analyses, we find our experimental settings adequate. But we certainly  acknowledge that expanding the models and settings remains a future research direction.
>
> > There should be an experiment on over-parameterized cases.
>
> The positioning of the computational experiments in our study aligns with the explanation provided above and holds meaningful implications even for experiments on MF networks that are not entirely over-parameterized. Particularly, differences in performance stemming from the representational power between MF networks and NTK networks are evident and verifiable even within finite-width MF networks. Further details regarding this point are directed to the response for reviewer GTpe (Benefits of feature learning in the context of this RL).
>
> > Why are the networks over-parameterized?
>
> The use of over-parameterized neural networks, such as those in mean-field theory and NTK, does not derive from the motivation that increasing the number of network parameters inherently improves performance. Over-parameterization is an approach aimed at providing theoretical guarantees for neural network optimization and is driven by demands from the theoretical side. Therefore, while achieving theoretical guarantees with non-over-parameterized finite-width MF networks would be an ideal goal, the analysis of neural networks remains incomplete even on supervised learning, particularly in the context of reinforcement learning, where analyses often remain limited to linear function approximation like NTK or more powerful assumptions. Hence, the deep learning theory community is actively exploring statistical and optimization properties beyond NTK. Our research contributes as a significant step within this array of research directions in reinforcement learning.
>
> (continued below)

---

> > ### Author Response · Authors · 2023-11-22
> > **Rebuttal by Authors**
> >
> > (continued)
> >
> > **Regarding Feature Learning in this Work**
> >
> > > I would like to see the details of the feature learning in this work. Is there any encoded representation, like OFE and DREAMER?
> >
> > The concept of "features" addressed in our study does not directly refer to the "features" dealt with in OFE. When we mention "feature learning" here, we are highlighting the alignment of the first-layer bases with the data, which allows more efficient learning of the target function compared to methods like kernel-based techniques or linear operators. This becomes evident in the context of the curse of dimensionality. It is known that certain classes of functions, based on their smoothness across dimensions or low-dimensional manifold structures, evade the curse of dimensionality when neural networks perform regression tasks. Real-world functions typically reside within specific function classes that can bypass the curse of dimensionality, and practically implemented neural networks are observed to be less constrained by dimensionality. Therefore, the deep learning theory community refers to this as "feature learning," attributing this quality of neural networks over others like kernel methods. Although deep reinforcement learning, particularly Actor-Critic-based approaches, has seen significant success, our study positions itself as an understanding step by theoretically analyzing the learning dynamics of Actor-critic focusing on "feature learning."
> >
> > For a more detailed explanation regarding the significance of feature learning addressed in our study, please refer to our response to reviewer GTpe.

---

> > > ### Comment · Reviewer_1VQp · 2023-11-22
> > >
> > > Thanks for your responses which have addressed my concerns. I will increase my score to 5.
> > > I suggest the authors add the detailed introduction of mean field regimes and over-parameterized networks.
> > > They are important concepts in this paper.

---

### Official Review · Reviewer_fWZz · 2023-10-22

**Soundness:** 3 good
**Presentation:** 3 good
**Contribution:** 2 fair
**Rating:** 5
**Confidence:** 5

**Summary:**

The authors propose an actor-critic method to solve the RL problem in discrete time and continuous state and action space. Both the policy and the Q function are parametrized by neural network with one hidden layer. The parametrization becomes distributions in the mean-field limit.
The actor part is policy gradient w.r.t. regularized objective function, resulting in an Langevin dynamic of the actor distribution. The critic is temporal difference learning with two-level optimization. The inner loop is mean-field Langevin dynamic for q, while the out loop updates the critic objective.
The authors give proof for the convergence of the critic and whole algorithm in theorem 1 and 2 respectively. A numerical example is also provided to justify the algorithm.

**Strengths:**

The authors propose a new method to solve the RL problem and provide theoretical guarantee on the level of mean-field limit of neural network parametrization. This work contributes to enhancing our comprehension of deep learning techniques in solving RL problems.
The authors give a global convergence result for the algorithm.

**Weaknesses:**

Assumption 5 looks like a key part to fill the gap of the proof. But there is neither intuitive explanation, nor examples showing the validity of this. I am concerned about the existence of such a uniform positive lower bound 1/B. Given that the policy \pi belongs to a rich family, the advantage functions A_{\pi} should also span a rich spectrum, making this assumption hard to satisfy.
The authors explain that when M and B are large, the class \mathcal{F}_{R,M} is a wide class. This argument is fine for M, but for B I do not understand. It is unclear to me why Assumption 5 constitutes a constraint on the class \mathcal{F}_{R,M} and how increasing the value of B contributes positively. In my understanding, a larger class \mathcal{F}_{R,M} might increase the likelihood of satisfying this assumption. Also, this explanation does not justify the existence of a uniform positive lower bound 1/B.

**Questions:**

1.	Page 3. \varrho_\pi is the stationary distribution, while \nu_\pi is state visitation measure. I think the difference is that the latter depends on the initial distribution. If the initial distribution is the stationary distribution, then they should be the same, in which case the definition of \nu_\pi is unnecessary. However, the authors are using the policy gradient theorem, which only holds under stationary distribution.
2.	Page 4 SxA should be in R^(d-2) with d>3? There is 1 dimension for b and another one for the bias of the network.
3.	In the algorithm, is [0,m] a common notation for 1 to m? I think (0,m] makes more sense.
4.	Page 5. What’s the meaning of “its minimum value always upper bounds the mean squared error”?
5.	Page 6 upper part. There are two definitions for \mathcal{L}_l[q] and they do not coincide with each other. Maybe the second one should be modified.
6.	Lemma 1 eqn (9). The second term on the right looks confusing. Q^{(l+1)} is a function of (a,s) while q_*^{(l+1)} is a distribution with input \omega. Their difference doesn’t make sense. Please clarify this part.
7.	Page 7 bottom. What is the meaning of “can achieve the annealed Langevin dynamics by attenuating \lambda_{TD} by O(1/log(S))”?
8.	Page 2 bottom. In related works, when introducing the LQR settings, I think the work “Single Timescale Actor-Critic Method to Solve the Linear Quadratic Regulator with Convergence Guarantees” (JMLR2023) is also closely related.
9.	A notation issue: s is used for both state and time, which is a bit confusing. t is used for both the discrete time in RL and the continuous Langevin dynamic.
10.	The dynamic for critic for the outer loop is discrete. The continuous dynamic is only for inner loop. We only have Q^{(l)}, and Q_t is not explicitly defined. I think it is better to give an explicit definition of Q_t based on the inner loop and clarify (at Lemma 2) that at some points of t, it may not be differentiable.

There are also some typos:
Page 1: a considerable challenge “to” the optimization aspect
Page 5 after (6). g[\rho_t] should be g_t[\rho_t]?
Page 6 eqn (7). \mathcal{T} should be \mathcal{T}^\pi?
Page 8 Theorem 2. Let J* be “the” optimal expected total reward.

---

> ### Author Response · Authors · 2023-11-22
> **Rebuttal by Authors**
>
> Thank you for your helpful comments. We address the technical comments below.
>
> **Regarding Assumption 5; some practical example**
>
> Thank you for your insightful remarks regarding Assumption 5 in our manuscript. Indeed, Assumption 5 plays a crucial role in bridging the gap within the proof. Sufficient increase of the parameters $R$ and $M$ ensures the richness of $\mathcal{F}\_{R, M}$. On the other hand, it's worthing to note that Assumption 5 guarantees that when one has a policy, one can always approximate the advantage function in the gradient direction of the policy gradient within the finite KL-divergence ball. Indeed, for example, Assumption 5 is satisfied when $A_{\pi}/\|A_{\pi}\|\_{\sigma_{\pi}}\in \mathcal{F}\_{R, M}$.
> Now that $Q_{\pi}\in\mathcal{F}\_{R, M}$ is assumed by Assumption 2, the geometric regularity of Assumption 5, coupled with the richness of the function class $\mathcal{F}\_{R, M}$, is a further generalization of the above example $A_{\pi}/\|A_{\pi}\|\_{\sigma_{\pi}}\in \mathcal{F}\_{R, M}$, with the assumption relaxed by an amount corresponding to $B>0$.
>
> **Addressing questions:**
>
> I appreciate your invaluable feedback, which prompted several clarifications and adjustments in our manuscript:
>
> 1. Please see Theorem 1 in Sutton et al. (1999). Regarding the definition of $d^{\pi}$ in Sutton et al. (1999)'s Theorem 1, it is indeed associated with the visitation distribution in their paper. While they initially define $d^\pi$ in the context of the stationary distribution, they later redefine it as the visitation distribution just before Theorem 1. This transition might cause confusion, but it is crucial to note that within Theorem 1, $d^{\pi}$ refers to the visitation distribution. This alignment is evident from the proof. Consequently, for our Proposition 1, similar consideration of expectations concerning the visitation distribution is necessary, leading to the current statement. Moreover, you are correct in stating that when the initial distribution is the stationary distribution, the visitation distribution coincides with the stationary distribution, naturally fulfilling one of the conditions in Assumption 4.
> 2. I have rectified the confusion regarding dimensions by modifying it to $\mathcal{S}\times\mathcal{A}\subset \mathbb{R}^{D}$.
> 3. The correction has been made to reflect that particles range from 1 to $m$: $[1,m]$.
> 4. Regarding the inner optimization objective function $\mathcal{L}\_l$ in MFLTD, it has been emphasized that this function is convex over the probability distribution space and that the values of the inner objective function at its stationary point is larger than a constant multiple of the true objective function, the expected squared error $\frac{1-\gamma}{2}\mathbb{E}{\varsigma_{\pi}}[(Q_q - Q_{\pi})^2]$, with an error of at most $\mathcal{O}(\lambda_{TD})$. This implies a continuous reduction of $\frac{1-\gamma}{2}\mathbb{E}_{\varsigma_{\pi}}[(Q_q - Q_{\pi})^2]$ with each inner update. To enhance clarity, I have revised the respective section for better comprehension.
> 5. As per your observation, I have rectified notation inconsistencies and modified the definition of the second $L_l[q]$.
> 6. The erroneous reference to $q_*^{(l+1)}$ instead of $Q_*^{(l+1)}$ has been corrected.
> 7. In our study's setup, although we initially maintained $\lambda_{TD}$ as a constant across all steps, your insight on adapting $\lambda_{TD}$ as a function of the inner optimization time $s\in[0, S]$, such as $\lambda_{TD} = 1/\log(s)$, offers a means to progressively decrease biases in the convergence values of the objective function. This adjustment aims to converge asymptotically towards the true optimal value.
> 8. Your suggestion regarding additional references is duly noted. I will include the relevant citation in the specified section.
> 9. Apologies for any confusion caused by overloaded notations. I have revised the notation system to a clearer and unambiguous format to mitigate any potential misunderstandings.
> 10. In MFLPG (Algorithm 1), in each policy evaluation step, we uniformly sample $l\in[T_{TD}]$ and adopt $Q^{(l)}$ as $Q_t$ from the estimated Q-functions $\{Q^{(l)}\}\_{l\in[T_{TD}]}$ obtained by MFLTD (Algorithm 1). Then the error in the value function using $Q_t$ in the sense of expected value is upper-bounded by the error on the right-hand side of Theorem 1. Supplementary explanation is added in the paper.

---

> ### Comment · Reviewer_fWZz · 2023-11-22
> **Thank you for the explanation, the paper looks clearer.**
>
> For assumption 5, do you mean the choice of f could depend on $\pi$? If this is the case, then please clarify. What I understand from Assumption 5 was that the choice of f is independent of $\pi$.
> Question 1 was my mistake, thank you for the clarification.

---

> > ### Author Response · Authors · 2023-11-22
> > **Answer from the authors to your additional questions**
> >
> > Thank you for your prompt reply.
> > Yes, that's correct. The selection of  $f$ indeed depends on $\pi$. Given the assurance of $Q_{\pi}, \pi \in \mathcal{F}\_{F,R}$ at this stage, the existence of this \$A_{\pi}/\|A_{\pi}\| \in \mathcal{F}\_{R, M}$ is generally expected. Similar assumptions have been utilized, for instance, in the analysis of policy gradient using NTK as seen in Wang et al. (2019).
> >
> > Feel free to make any adjustments as needed to better align with your context or style preferences.
> >
> > Best regards,

---

### Official Review · Reviewer_GTpe · 2023-11-03

**Soundness:** 3 good
**Presentation:** 2 fair
**Contribution:** 2 fair
**Rating:** 5
**Confidence:** 3

**Summary:**

The work presents a convergence theorem for neural actor-critic algorithms with feature learning (i.e. in the mean feild setting). They do so by formulating neural networks as a collection of $m$ neurons and consider the mean filed behavior as $m \to \infty$ under a pecific parametrization of the single hidden layer NNs. Both the actor and critic NNs are parameterized similarly. The agent is assumed to be optimizing an entropy regularized objective and optimized via minimizing a surrogate first variation. They employ a dual loop learning mechanism for the $Q$ function, within every policy improvement step. They show the soundness of their methodology:
1. they show "one step" improvement of the inner loop of TD learning,
2. convergence of the $Q$-function's estimate to the $Q$-function corresponding to the policy in that step
3. convergence to globally optimal stationary distribution with bias dependent on the entropy regularization's weight.

Finally, they demonstrate the efficacy of their proposed method in a toy environment.

**Strengths:**

The paper has the following strengths:

1. The authors clearly state and back each one of their assumptions.
2. The authors build the algorithm and the theoretical framework for analysis in a structured manner.
3. They closely and clearly follow previous work on neural TD learning for NTK parametrization (Cai et. al 2019 and Zhang et. al 2020) while formulating the problem.
4. The overall results and conclusion are pretty convincing (although I am not certain of all the proofs and details).

**Weaknesses:**

I see a couple of weaknesses:

1. **Weaknesses in writing:**  there are issues with notation and referencing sections of appendix. As a reader I would be interested in appendix sections with various proofs and technical concepts. For more details see the minor issues listed below. Further, there is some lack of clarity when it comes to interpreting the various Lemmas and Theorems. There are some notational issues as well which make it harder for me to interpret the math and the results.
2.  **Issues with feature learning in RL:** while the authors claim that the NN learns features they do not define it in context of RL. I understand that mean field parametrization is a feature learning regime as opposed to NTK but I dont see how it relates to RL? In section 5, the claim is that MFLTD performs better by reducing the Bellman error better than NTK-TD due to better feature learning but I am unclear how it would follow that feature learning helps achieve this. What contribution does feature learning make to agent performance or to minimizing the TD error? Further, this is only shown for the td-learning sub-loop (MFLTD) and not for the broader algorithm MFLPG. Any insight into how feature learning effects an RL agent's learning trajectory would be meaningful and go a long way in answering the primary question posed in the introduction.

***Minor issues:***
1. Agarwal et. al on top of page 3 is missing the year.
2. Section 3.1 for Log-Sobolev inequality point to Appendix and please offer some explanation.
3. Section 3.1: please cite previous work for the claim that regularization smoothens the problem
4. "is a standard Gaussian distribution simply." -> "is a standard Gaussian distribution."
5. Point to the proof of Proposition 1 in Appendix and also the related Appendix sub-section which contains the definition for First variation of Functionals.
5. Section 3.1: cite for the connection between the Fokker Planck equation and the SDE.
6. Algorithm 2 is referenced in algorithm 1 without any explanation, I would add one liner on how Algorithm 2 is for TD-learning.
7. Definition 1 in Appendix B.1: What is $\mathcal F$? It would be better defined or explained here.
8. $t$ is overloaded across Equations 5,6 and Section 2. One is agent time and another is gradient time step.
9. What do you mean by "From an argument similar to $\mathcal F$" in section 3.2?
10. Term $s$ is overloaded for the expression of $dw_s$ for Section 3.2, used as both state and time.
11. In the sentence with sample $(s, a, s', a')$ is introduced without explanation, where are these samples used? In the estimation of the loss in Equation (7)?
12. Equation (25) from Appendix referenced in the main body in Section 4 and this seems like a leap.
13. $S$ as run-time is overloaded in Section 4.
14. Lemma 1: what is the difference between $\varsigma$ and $\varsigma_{\pi}$?
15. In various locations $Q_{\pi}$ and $Q_q$ are used which is ambiguous notation because $\pi$ is policy and $q$ is the initialization distribution of the NN.
16. Theorem 1: $s$ is overloaded.
17. Section 4: "takes advantage of the data-dependent advantage of neural networks" reads like it might be wrong.
18. It might be helpful to define the Radon-Nikodym Derivative in the Appendix.
19. Figure 1 left side: why is the x-axis starting from 1000?
20. Point to proof sections in the Appendix under all statements of Theorems and Lemmas.

**Questions:**

See above.

---

> ### Author Response · Authors · 2023-11-22
> **Rebuttal by Authors**
>
> We sincerely appreciate your time and effort in reviewing our manuscript and providing valuable feedback. We have taken them into serious consideration for improving the clarity and coherence of our work.
>
> > Benefits of feature learning in the context of this RL
>
> As you rightfully pointed out, the Mean-field (MF) scaling approach utilized in our study, in comparison to the Neural Tangent Kernel (NTK) scaling prevalent in existing research, covers a more diverse function class by aligning features along the data used for learning. In Section 5, we performed a comparison between NTK scaling and MF scaling using finite-width three-layer neural networks. Admittedly, extrapolation from the analysis results of our study focused on infinite-width neural networks may require further consideration in this section. By employing finite-width MF networks, we aim to elucidate the benefits of feature learning, specifically highlighting three aspects: (1) the performance difference stemming from the representational capabilities of MFLTD and NTK-TD, (2) Sample complexity with respect to the dimension, (3) Reinforcement learning unique feature learning trends:
>
> **(1) The performance difference due to representational capabilities.**
>
> - Primarily, NTK scaling employs fixed feature bases throughout training, only learning the second layer based on these bases, and its expressive power relies on the number of available bases determined by the network's width. In contrast, in MF scaling, the bases corresponding to the first layer's features can adequately adapt to the data used for learning.
> - As a specific example, to represent the polynomials of few relevant dimensions in a high (, say $d$-,) dimensional space, superpolynomial width is required in the NTK scaling because with high probability the first layer weight of each neuron does not align with such relevant directions. On the other hand, for the MF scaling, the required width does not depend on the ambient dimension $d$ (although here we ignored an optimization aspect). Detailed discussion can be found in [Damian et al. (2022)](https://arxiv.org/pdf/2206.15144.pdf).Like this, in terms of representation ability, the function class realized by finite-width NTK networks is sometimes clearly distinguished from that by the NTK limit, and less able to approximate functions common in learning theory than those by MF networks with finite width (although the function class of finite-width NTK networks are not completely included in that of finite-width MF networks). Therefore, the class of finite width networks in [Cai et al. (2019)](https://proceedings.neurips.cc/paper/2019/hash/98baeb82b676b662e12a7af8ad9212f6-Abstract.html) is not as flexible as that of finite width MF networks.
> - We wanted to emphasize not only the order itself but also the definition is important to discuss the *convergence rates*. The measurement of convergence depends on with which the MSE is measured (, which we called *global minima*). For a fixed finite width, the global minima in MF can be significantly closer to the true function than that in NTK as we argued above. Although our network is infinite width, this is mainly not for representation but for optimization, and it would be possible to convert our results to the finite width setting.
>
> **(2) Sample complexity with respect to the dimension.**
>
> The MF network with infinite width also has advantage over the NTK network with infinite width, in terms of sample complexity in high-dimensional problems.
>
> - If you look at the learnt network in MF, the first layer parameters typically align with the important directions. But not in NTK, because NTK weight does not allowed to travel so much.
> - As a result, the MF learnt network typically has low-dimensionality, and therefore requires fewer samples. Specific examples are [parity function](https://openreview.net/pdf?id=swEskiem99), [polynomials of few relevant dimensions](https://arxiv.org/abs/2206.15144), [low dimensionality](https://openreview.net/forum?id=6taykzqcPD), [hierarchical functions](https://arxiv.org/abs/2302.11055), and [single index models](https://proceedings.neurips.cc/paper_files/paper/2022/hash/f7e7fabd73b3df96c54a320862afcb78-Abstract-Conference.html). For example, in order to learn the $2$-parity in a $d$-dimensional space, $\mathcal{O}(d^2/\varepsilon)$ sample is required for NTK but $\mathcal{O}(d/\varepsilon)$ for MF ([Telgarsky (2023)](https://openreview.net/pdf?id=swEskiem99)) to achieve the accuracy. However, the dimension dependency will be absorbed by just looking $T$ or $T_{TD}$.
> - This is not because MF has less (or more) representation ability and the whole hypothesis class is smaller (or larger), but because the parameters can gain the aligned structure a result of training (=feature learning).
>
> (continued below)

---

> > ### Author Response · Authors · 2023-11-22
> > **Rebuttal by Authors**
> >
> > (continued)
> >
> > **(3) Reinforcement learning unique feature learning trends.**
> >
> > - Feature learning is very useful in the context of RL. In particular, RL agents often encounter state-action samples possessing low-dimensional structures. For instance, when training a reinforcement learning model to control a robotic arm, the feasible states are not uniformly distributed across the entire space but are constrained to specific subspaces due to their mechanical limitations. Additionally, in cases where images are used as state inputs, the learning data can often be projected onto lower-dimensional spaces. This reduction in dimensionality plays a crucial role not only in the implicit learning within NN but is also explicitly addressed in some recent studies ([Dayan 1993](https://ieeexplore.ieee.org/document/6795455), [Yang et al. 2021](https://openreview.net/pdf?id=edJ_HipawCa), [Huang et al. 2022](https://arxiv.org/abs/2110.05721), [Lan et al. 2022](https://proceedings.mlr.press/v151/le-lan22a/le-lan22a.pdf)).
> > - In response to these research trends, recent studies ([Zhang et al. 2020](https://proceedings.neurips.cc/paper/2020/file/e3bc4e7f243ebc05d66a0568a3331966-Paper.pdf), [Zhang et al. 2021](https://proceedings.neurips.cc/paper/2021/file/85a4413ecea7122bcc399cf0a53bba26-Paper.pdf)) confront the implicit representation learning optimization in neural networks through the utilization of Lazy-training. On the other hand, our study aims to revisit the significance of MF networks in this context. Through our research, we expect to demonstrate how the efficiency of representation learning, already established in supervised learning paradigms as mentioned earlier, can similarly benefit reinforcement learning.
> >
> > In summary, it has become a "common sense" in the community that the mean field regime provides more flexible approximation and adaptivity due to its feature learning ability so that it yields better generalization. The community of deep learning theory is now extensively investigating statistical and optimization properties in "beyond-NTK" regime. Our study provides an important step in the series of such a research direction for reinforcement learning.
> >
> > I have commented below on some of the minor issues you have raised that need to be supplemented.
> >
> > > 8. What is $\mathcal{F}$?
> >
> > The $\mathcal{F}$ was a mistake for $F$.
> >
> > > 10. What do you mean… ?
> >
> > $L_l$ was a notational error that has been corrected.
> >
> > > 18. Section 4: "takes advantage of the data-dependent advantage of neural networks" reads like it might be wrong.
> >
> > Analyses of TD learning that rely on lazy-scaling and use function approximations of all existing NNs have limited learning ability due to the initial value dependence of the representation basis of the NNs, regardless of the data. In contrast, this study follows MF-scaling, and the analysis results show that the parameters of all NNs depend on the input data for efficient learning, capturing the inherent goodness of the NNs.
> >
> > > 20. Figure 1 left side: why is the x-axis starting from 1000?
> >
> > The purpose of the experiment shown in Fig. 1 was to determine how much difference in expressive power actually occurs when NNs are trapped in the NTK regime, as opposed to the MFLTD proposed in this study. For this reason, the first half of the experiment, which provides little information, was omitted to focus on the results of the convergence point, and to avoid making the figure too small due to the scale by including the first half, which has a large vertical change.

---

### Official Review · Reviewer_XKHS · 2023-11-04

**Soundness:** 3 good
**Presentation:** 3 good
**Contribution:** 2 fair
**Rating:** 5
**Confidence:** 3

**Summary:**

This work proposes actor-critic methods where both the actor and critic are represented by over-parameterized neural networks in the mean-field regime as shown in Eq. (3). They proposed entropy and $\ell_2$ norm regularized expected reward objective, and its policy gradient update (continuous and discrete-time),  as shown in Algorithm 1. The critic update contains two loops (to guarantee monotonic improvement), as shown in Algorithm 2.

Under several assumptions, the authors proved that Algorithm 2 converges in terms of Q-function with rate $1/T$, verifying its validness as a policy evaluation algorithm, and then Theorem 2 shows that Algorithm 1 converges linearly toward globally optimal policy value up to bias introduced by regularization. Numerical results verify the theoretical findings.

**Strengths:**

1. Actor-critic methods with neural network parameterizations and theoretical guarantees give promising results, justifying their empirical success.
2. The writing and presenting results are clear and easy to follow.

**Weaknesses:**

1. Hard to verify assumptions, which weaken the aim of supporting empirical success by theoretical footing.
2. It is unclear how practical those proposed actor-critic methods are. The authors argued that those assumptions are moderate, but the work also does not show the proposed methods are still close enough to what have been used in practice.

**Questions:**

How do we justify widely used actor-critic methods in practice are learning representations in the same way of how the proposed methods learn?

---

> ### Author Response · Authors · 2023-11-22
> **Rebuttal by Authors**
>
> Thank you for your helpful comments. We address the technical comments below.
>
> > Hard to verify assumptions, which weaken the aim of supporting empirical success by theoretical footing.
> >
> > It is unclear how practical those proposed actor-critic methods are. The authors argued that those assumptions are moderate, but the work also does not show the proposed methods are still close enough to what have been used in practice.
>
> **Supplementary Information on Assumptions' Validity and Limitations**
>
> Each assumption in our study aligns closely with the commonly used assumptions in theoretical analyses of neural reinforcement learning. They are quite familiar within this domain.
>
> Assumptions 1 and 3 effectively dictate the richness of the neural network function class induced by determining the activation functions within neural networks. Our coverage encompasses a wide range of neural networks employing typical activations, such as sigmoid or hyperbolic tangent functions applied component-wise. These assumptions are prevalent in other analytical studies, like Agazzi & Lu (2021), [Leathy et al. (2022)](https://proceedings.mlr.press/v162/leahy22a/leahy22a.pdf), and Zhang et al. (2021, 2022). Additionally, Assumption 1 is validated in cases where kernels are smooth and light-tailed, like the RBF kernel as mentioned in [Suzuki et al. (2023)](https://arxiv.org/abs/2306.07221) for applications such as MMD and KSD estimation.
> Assumption 3 naturally arises from the richness of the neural network class introduced by Assumption 1. As detailed in Appendix B.1, this is an inherent deduction from the benefits of the Barron class' universal function approximation capability, among other factors.
>
> Assumption 5 ensures that when a policy is present, approximating the advantage function within a finite KL-divergence ball in the gradient direction of the policy gradient is always possible. Specifically, for instance, when $A_{\pi}/\|A_{\pi}\|\_{\sigma_{\pi}}\in \mathcal{F}\_{R, M}$. Given $Q_{\pi}\in\mathcal{F}\_{R, M}$ by Assumption 3, the moderate geometric regularity of this assumption, coupled with the richness of the function class $\mathcal{F}\_{R, M}$, is evident. This assumption is a relaxed version accommodating $B>0$, as elaborated in the mentioned example.
>
> We've supplemented and complemented the main text with additional clarifications, realistic examples of meeting these assumptions, and more. If there are further questions or doubts regarding specific assumptions' limitations or validity, please feel free to inquire.
>
> **Insights on the Practicality of Proposed Algorithms**
>
> Section 5 of our study offers valuable insights into the practicality of our proposed algorithms. For instance, our experiments demonstrate that while MFLTD constitutes a double-loop algorithm, it performs comparably to a general TD(1), displaying even higher performance accuracy. This emphasizes the practical adequacy of our algorithm in obtaining more accurate Q-functions.
>
> > How do we justify widely used actor-critic methods in practice are learning representations in the same way of how the proposed methods learn?
>
> "Representation learning" is essentially the advantage of using neural networks as learning machines themselves. In contrast to kernel methods, multi-layered neural networks (having two or more trainable layers) achieve more efficient learning by aligning features based on the data used for learning. In our study, we capture the learning dynamics of widely used neural network representations through the Mean-Field (MF) regime. This understanding aligns with a firmly established "common knowledge" within the community, especially within the context of general supervised learning, where learning dynamics of neural networks in the MF regime partially mirror those utilized in practical scenarios.
>
> For instance, considering MFLPG as policy gradient where Gaussian noise is added each gradient update in the discrete-time algorithm, this noise inclusion leads to an optimization under Langevin dynamics, ensuring analytical proof. That is, this addition does not fundamentally alter the learning of network representations from a viewpoint of representation learning. Consequently, real-world neural networks inherently possess a learning mechanism quite similar to those handled in our study. This fact contributes to supporting the assertion that widely used actor-critic methods in practice learn representations in a manner akin to the proposed methods.
>
> The learning of "representations" in the implementation of individual reinforcement learning algorithms, while crucial, is not straightforward to elucidate. There remains ample scope for further, more rigorous research into the specific implications of implicit "representation learning" through neural networks within the realm of reinforcement learning. For more insight into neural network representation capabilities, please refer to the response to reviewer GTpe.

---

### Official Review · Reviewer_Xd7S · 2023-11-07

**Soundness:** 2 fair
**Presentation:** 2 fair
**Contribution:** 3 good
**Rating:** 6
**Confidence:** 3

**Summary:**

This paper considers the mean-field neural actor-critic algorithm for deep reinforcement learning. Both policy and Q-function are parametrized by a two-layer neural network. With appropriate assumptions, the authors conduct analysis in the mean-field limit region. The main contributions in this paper are
1. Authors introduce a mean-field Langevin TD learning algorithm for the critic update. It is a double-looped algorithm, and specifically the inner loop is based on the gradient descent of the regularized loss function discussed in Nitanda et al., 2022 and Chizat, 2022.  For the discrete time analysis, the paper provides a time-averaged convergence rate $O(1/T)$.
2. Authors introduce a mean-field Langevin policy gradient algorithm for the actor update. From the continuous-time perspective, the paper provides the guarantee of global optimality and a linear convergence rate.

**Strengths:**

The algorithms for the actor and critic updates combined with feature learning are original, and rigorous analysis is provided. Moreover, the authors claim that the paper gives the first global optimality result of the stationary point of the MFLPG using a one-point convexity method. Numerically the paper shows that it outperforms NTK and single loop TD algorithm by achieving lower mean squared Bellman errors.

**Weaknesses:**

1. The writing of this paper is very dense and the text arrangement makes it not easy to track definitions and theorems. I suggest that all equations in the main text should be labeled, and for notations in the appendix, authors should try to make an enumerated list to summarize all definitions of $\tilde{f}$, $\tilde{\rho_t}$, $\hat{\rho_t}$, $Q^{(l)}$, $Q_{\pi}$, $q_s$, $q^{(l)}$, $\cdots$ to assist reading.

2. The paper does not show how the actor and critic updates are combined together. A unified analysis for the combined algorithm would also be useful. However, I cannot find the combined algorithm anywhere in this paper.

3. The discrete time analysis of MFLPG, unlike MFLTD, is missing.

4. Many parts of the proof are unclear to me, I will explain them in details in the Questions section.

5. The technique used to show linear convergence of inner loop MFLD does not seem to be new compared to previous established works.

**Questions:**

In the appendix,

1. Is $||f||\{\mu,2\}$ the same as $\mathbb{E}_{\mu}[f^2]^{1/2}$?

2. On page 16, in C.1, (12), where is $\frac{\lambda_{TD}}{2}\mathbb{E}(||\omega||^2) +Z$ ?. Also, $s\in [0,T_{TD}] $ should be $l\in[0,T_{TD}]$.

3. On page 16, $\mathbb{E}[(Q^{(l)}-\mathcal{T}Q^{(l)})(Q^{(l+1)}-Q_\pi)] = \mathbb{E}[\Delta Q^{(l+1)}(I-\gamma \mathcal{P})\Delta Q^{(l)}]$, can you give a detailed derivation?

4. On page 18, ``using the strong convexity of $L_l$", can you explain or add a reference to it?


5. On page 19, the explanation of $D_{KL}(q_0||\nu)=0$ is missing.

6. On page 21, the expression of $J[\rho]$ is different from the one on page 3, can you add details to explain why they are the same?

7. On page 22, $g_t[\rho_t]$ does not look like the one defined on page 5, (5). A derivation for that equivalence is missing.

8. On page 22, (31), I don't see how the last inequality holds. In 1D, it means that $2ab\geq a^2-b^2$, how could it be true?

In the main text,

9. On page 4, $\mathcal{S}\times \mathcal{A}\subset \mathbb{R}^d$ does not look correct.

10. Assumption 2 needs a further verification. What do you mean by "$R$ is the boundary of neural networks Q-function estimator"? Or say ``neural network radius" mentioned on page 20? I suggest that giving a simple example and computation would make this assumption more convincing.

---

> ### Author Response · Authors · 2023-11-22
> **Rebuttal by Authors**
>
> I appreciate your thorough evaluation of our work and have made specific adjustments based on your invaluable suggestions:
>
> > authors should try to make an enumerated list to summarize all definitions…
>
> I am sincerely grateful for your feedback. In line with your recommendation, I intend to enhance the readability of the manuscript by compiling a supplementary document summarizing essential symbols used in proofs and theorems.
>
> > The paper does not show how the actor and critic updates are combined together.
>
> We would like to emphasize that our main result Theorem 2 evaluate the errors induced by both algorithms by combining the error analysis for each algorithm.
> Indeed, you can find that the MFLAC in Algorithm 1 entails running the policy evaluation (MFLTD, Algorithm 2) at each update of policy improvement.
> Then, the error given by Theorem 2 yields the statement under the condition of $T_{\mathrm{TD}} = O(1/\lambda_{\mathrm{TD}})$ for the number of iterations in the inner loop.
>
> > The discrete time analysis of MFLPG, unlike MFLTD, is missing.
> >
>
> Our research analyzed the continuous-time dynamics of MFLPG. Expanding this analysis to discrete-time dynamics can naturally extend from the continuous-time framework. This extension, dependent on the learning rate $\eta>0$, converges toward the optimal policy reward within an error margin, as observed in recent studies on time and particle discretization (Suzuki et al. (2023)).
>
> > The technique used to show linear convergence of inner loop MFLD does not seem to be new compared to previous established works.
>
> **Impact on the paradigm of analysis of TD learning of NNs in MFLTD**
>
> Acknowledging your point, the uniqueness of stationary points in the inner loop of MFLTD and its global optimality leverage previously established convergence proofs. However, the contribution of MFLTD in our study lies in proposing this dual-loop algorithm and deriving non-asymptotic results regarding the convergence of its outer loop. As described on page 5, introducing a naive semi-gradient method using gradient descent on the optimal problem in the mean-field regime, i.e., as an optimal problem on the probability distribution space, cannot be achieved. Specifically, the Fokker-Planck equation $\partial_t \rho_t = \lambda\cdot\Delta \rho_t + \nabla\cdot\left(\rho_t \cdot v_t\right)$, where $v_t := \nabla\mathbb{E}{\varsigma_{\pi}}[(Q_q-\mathcal{T}^{\pi}Q_q)h_{\theta}]$, does not converge. This is due to the fact that the naive semi-gradient on the probability subspace $\mathbb{E}{\varsigma_{\pi}}[(Q_q-\mathcal{T}^{\pi}Q_q)h_{\theta}]$ does not consistently align with the true oracle gradient direction $\mathbb{E}{\varsigma_{\pi}}[(Q_q-Q_{\pi})h_{\theta}]$. This discrepancy opposes the intuition derived from the demonstration in Cai et al. (2019), where, in naive vector space TD learning optimization with linear function approximation of the Q-function, the semi-gradient $\mathbb{E}{\varsigma_{\pi}}[(Q_{\Theta}-\mathcal{T}^{\pi}Q_{\Theta})\cdot\nabla_{\Theta} Q_{\Theta}]$ aligns with the oracle gradient direction $\mathbb{E}{\varsigma_{\pi}}[(Q_{\Theta}-Q_{\pi})\cdot\nabla_{\Theta} Q_{\Theta}]$. Hence, to estimate the true function $Q_{\pi}$, we propose a novel TD learning algorithm using the direction of the TD error $\mathbb{E}{\varsigma_{\pi}}[(Q-\mathcal{T}^{\pi}Q)h_{\theta}]$. This proximal dual-loop algorithm goes beyond the analysis of existing NTK scaling, achieving probabilistic distribution optimization and demonstrating convergence in MF networks, contrary to the aforementioned intuitive discrepancy. Our study provides the first convergence analysis of TD learning in MF networks without great regularization (e.x. using sufficiently large regularization parameter $\lambda>$const. in Leahy et al. 2022). Additionally, the practical validation of this newly proposed algorithm in Section 5 is a noteworthy contribution as well.

---

> > ### Author Response · Authors · 2023-11-22
> > **Rebuttal by Authors**
> >
> > **Addressing the questions**
> >
> > Here are the clarifications and revisions we've made in response to your queries:
> >
> > 1. It is correct. Additional notations and their explanations have been included in Appendix A for reference.
> > 2. The range should indeed be $l\in[0, T_{TD}]$, and we made the necessary correction. Moreover, regarding the former question, it holds that$\frac{\lambda_{TD}}{2}\mathbb{E}[\|\omega\|^2] + \lambda_{TD}\mathrm{Ent}[q] \propto \lambda_{TD}D_{\mathrm{KL}}(q\|\nu)$, which results from $\nu\sim\mathcal{N}(0, 1)$.
> > 3. For the equation $\mathbb{E}{\varsigma_{\pi}}[(Q^{(l)}-\mathcal{T}Q^{(l)})\cdot(Q^{(l+1)}-Q_{\pi})] = \mathbb{E}{\varsigma_{\pi}}[\Delta Q^{(l+1)}(I-\gamma \mathcal{P}) \Delta Q^{(l)}]$, the following transformations are deduced: $$\mathbb{E}{\varsigma_{\pi}}[(Q^{(l)}-\mathcal{T}Q^{(l)})\cdot(Q^{(l+1)}-Q_{\pi})] = \mathbb{E}{\varsigma_{\pi}}[(Q^{(l)}-r-\gamma \mathcal{P} Q^{(l)})\cdot(Q^{(l+1)}-Q_{\pi})] = \mathbb{E}{\varsigma_{\pi}}[((Q^{(l)}-r-\gamma \mathcal{P} Q^{(l)}) - (Q_{\pi}-r-\gamma \mathcal{P} Q_{\pi}))\cdot(Q^{(l+1)}-Q_{\pi})] $$ $$= \mathbb{E}{\varsigma_{\pi}}[((Q^{(l)}- Q_{\pi})-\gamma \mathcal{P} (Q^{(l)}- Q_{\pi}))\cdot(Q^{(l+1)}-Q_{\pi})] = \mathbb{E}{\varsigma_{\pi}}[(I-\gamma \mathcal{P})(Q^{(l)}- Q_{\pi})\cdot(Q^{(l+1)}-Q_{\pi})] = \mathbb{E}{\varsigma_{\pi}}[\Delta Q^{(l+1)}(I-\gamma \mathcal{P}) \Delta Q^{(l)}],$$ where, for simplicity, we define that $\Delta Q^{(l)} = Q^{(l)} - Q_{\pi}$, $I$ is an identity operator, and $\mathcal{P}: L^2(\varsigma_{\pi})(\mathcal{S}\times\mathcal{A})\rightarrow L^2(\varsigma_{\pi})(\mathcal{S}\times\mathcal{A})$ as the linear operator such that $\mathcal{P} Q(s, a) = \int \mathrm{d} s' P(s'|s, a)\int\mathrm{d} a' \pi(a'|s') Q(s, a),\ Q\in L^2(\varsigma_{\pi})(\mathcal{S}\times\mathcal{A})$.
> > 4. I have supplemented an intuitive explanation for clarity. Apologies for any confusion caused. The sentence means that $L_l$ is merely a quadratic form when viewed as a functional of the probability distribution $q$, and that $\mathrm{Ent}[q]$ possesses convexity concerning the probability distribution $q$.
> > 5. As $q_0$ is defined as a Gaussian distribution, it follows that $D_{KL}(q_0\|\nu) = 0$.
> > 6. For simplicity, I denoted $J[\pi_{\rho}]$ as $J[\rho]$, as indicated on page 4. We corrected for clearer explanation.
> > 7. The consistent definition of $g_t$ is $g_t[\rho] = \mathbb{E}{\sigma_{\pi_{\rho}}}[A_{t}\cdot h_{\theta}] +\frac{\lambda}{2}\|\theta\|_2^2$. In accordance with the definition of $\tilde{\rho}_t = \exp\left(-\frac{1}{\lambda} g_t[\rho_t] - \ln\int\exp\left(-\frac{1}{\lambda} g_t[\rho_t]\right)\mathrm{d}\theta\right)$, it follows that $\partial_t \rho_t = \lambda\cdot\Delta \rho_t + \nabla\cdot\left(\rho_t \cdot\nabla g_t[\rho_t]\right) = \lambda\cdot\nabla\cdot\left(\rho_t \cdot\nabla \log\tfrac{\rho_t}{\tilde{\rho}_t}\right)$.
> > 8. Regarding the expression transformation, it adheres to $2ab\geq a^2 -(a-b)^2$. Note the distinction between $\tilde{\rho}_t$ and $\bar{\rho}_t$.
> > 9. The confusion between dimensions $d$ and $D$ has been rectified. We've corrected it to $\mathcal{S}\times\mathcal{A}\subset \mathbb{R}^{D}$.
> > 10. Apologies for any confusing expressions. Throughout our paper, we referred to the norm of the function class $\mathcal{F}\_{R, M}$ of neural networks as the "neural network radius" or "the boundary of neural networks Q-function estimator." Specifically, for a function class expressible by $\mathcal{F}_{R, M}$, the norm should be bounded by $R$. For instance, if the absolute norm of the Q-function is always bounded by $R_r$, the function class norm $R$ of the neural network estimating the Q-function must satisfy $R>R_r$.

---

### Meta-Review · Area_Chair_ixm6 · 2023-12-12

**Metareview:**

The manuscript analyzes the mean-field actor-critic algorithms beyond the lazy training regime. The convergence result establishes global optimality and feature learning. As pointed out by the expert reviewers, the presentation of the paper is quite dense and lacks some clarity. While the result is interesting, some of the theoretical assumptions are hard to justify. Overall, while the AE thinks this is an interesting paper, in its current shape it does not meet the standard of acceptance.

**Justification For Why Not Higher Score:**

see metareview.

**Justification For Why Not Lower Score:**

N/A

---

### Decision · Program_Chairs · 2024-01-16

Reject